# Comprehensive exploration of visual working memory mechanisms using large-scale behavioral experiment

Liqiang Huang 

Two decades of research on visual working memory have produced substantial yet fragmented knowledge. This study aims to integrate these findings into a cohesive framework. Drawing on a large-scale behavioral experiment involving 40 million responses to 10,000 color patterns, a quasi-comprehensive exploration model of visual working memory, termed QCE-VWM, is developed. Despite its significantly reduced complexity (57 parameters versus 30,796), QCE-VWM outperforms neural networks in data fitting. The model provides an integrative framework for understanding human visual working memory, incorporating a dozen mechanisms—some directly adopted from previous studies, some modified, and others newly identified. This work underscores the value of large-scale behavioral experiments in advancing comprehensive models of cognitive mechanisms.

The fusion of AI with extensive datasets has sparked a wave of pivotal studies that utilize vast internet data to investigate human behavior[1-6]. In the latest development of this trend, several recent research initiatives have begun conducting large-scale controlled experiments to gain deeper insights into basic cognitive processes. Each of these projects utilizes a large-scale experiment to carefully measure a specific facet of human behavior, subsequently conducts a "comprehensive exploration" of the relevant mechanisms/factors, and tries to build the most fitting model from them to interpret the accumulated data. This comprehensive exploration[7-9] approach attempts to combine the strengths of traditional experimental psychology with data-driven model development in the realm of artificial intelligence (AI). On one hand, this approach is theory-oriented: mirroring experimental psychology, it employs controlled experiments tailored to specific research objectives, striving to uncover theoretical insights. On the other hand, this approach is data-driven: similar to AI principles, this approach fundamentally relies on constructing an extensive, high-quality benchmark dataset as the cornerstone of model development, and iteratively refining the model to attain a satisfactory performance level. In brief, this is a theory-oriented, data-driven approach that uses AI tools (data-driven model development) to achieve the goal of experimental psychology (theoretical insights). A recent article provides a detailed conceptual and methodological justification for this comprehensive exploration approach[10].

For example, one pioneering study[7] merged existing theories of human decision-making through machine learning, yielding a model that provides predictions with superior accuracy compared to any singular theory. In a similar vein, another study[8] formulated a quasi-comprehensive exploration model for spatial working memory. This model, while remaining explicitly interpretable, approaches the accuracy of a convolutional neural network (CNN).

This comprehensive exploration approach offers a significant advantage over typical experimental psychology studies, which usually begin with a hypothesis and predict outcomes based on one or two pre-established dimensions. Increasingly, it is evident that fragmented insights from such narrowly focused studies cannot be easily synthesized into a cohesive overall picture[9-11]. The comprehensive exploration approach aims to address this limitation by creating an integrative framework that unifies these fragmented insights[10]. Naturally, unraveling the complex relationships between these fragmented mechanisms is statistically demanding, necessitating the use of large-scale experiments to provide sufficient data for robust analysis.

In this study, I embarked on a large-scale experiment to delve into the mechanisms underlying human visual working memory (VWM), a crucial domain for comprehending the complexities of the human mind[12-33]. Researchers have strived to delineate the intricate mechanisms inherent in VWM[34-37], but achieving a consolidated theory remains challenging. The dataset harnessed in this study is several hundred

Department of Psychology, The Chinese University of Hong Kong, Hong Kong, China. ✉e-mail: lqhuang@cuhk.edu.hk

times larger than those employed in previous investigations. This scale affords an avenue to formulate a comprehensive model that incorporates many established mechanisms and potentially reveals previously unreported ones.

In this study, participants carried out a delayed estimation task[15,38] related to VWM (See Fig. 1a and see the Methods section for more details). They were asked to memorize four saturated colors. After a 1-s retention interval, they were required to report each of the four colors by choosing it on a color wheel. The experiment employed a total of 10,000 randomly-generated color patterns and measured 40 million responses (1009 responses per item, SD = 32). As shown in Fig. 2, although the distributions of responses were generally centered around the presented colors, their shapes were notably distinct.

The study aims to develop a model that effectively explains the VWM process (i.e., fits the large-scale experiment data) while maintaining parsimony. Specifically, the model uses the 10,000 color patterns as input to predict the response distributions for 40,000 items (10,000 patterns × 4 colors). Two additional models are used for support: a baseline model representing previous theory-driven models and serving as the starting point, and a guidance neural network providing a benchmark for what a comprehensive model should achieve. Importantly, while the neural network plays a critical role as a reference tool, the target model itself is not an AI model; rather, it is a theory-based model similar to those traditionally used in VWM research[15–17].

Here, I show that a theory-based model—the QCE-VWM (described below)—simultaneously achieves effectiveness and parsimony. On one hand, the QCE-VWM is highly effective, outperforming neural networks in data fitting. On the other hand, it is fairly parsimonious, with only 57 parameters compared to the neural network's 30,796. The QCE-VWM provides an integrative framework for understanding VWM, incorporating a dozen mechanisms—some adopted directly from prior studies, others modified, and several newly identified.

## Results

### Pattern-level summary

The raw data are first summarized into response distributions for the 40,000 items (10,000 patterns × 4 colors) by amalgamating responses from all trials featuring the same pattern. These response distributions are then used in all subsequent modeling efforts. See Supplementary Methods 2.1 for reasoning behind this summarization.

### A neural network

A neural network was employed to analyze the dataset. As depicted in Fig. 1b, this neural network features an input layer with 8 nodes, which is fully connected to a first hidden layer comprising 100 neurons. After this layer, a ReLU activation function, defined as ReLU($x$) = max($x$, 0), is applied. This first hidden layer is fully connected to a second hidden layer that also contains 100 neurons, and again, a ReLU activation function is applied. Finally, the network possesses an output layer with 196 neurons.

The 8 input values are grouped into four pairs, each representing a color as the x/y coordinates on a color wheel. The 196 output values are designed to be used to simulate the distribution of responses. This neural network aims to emulate observers' responses by blending 16 normally distributed components - which represent knowledge-based responses - with a fraction of random guesses.

Subsequently, the distribution of responses is computed and compared against the ground truth (i.e., the actual distribution of observers' responses). The network was optimized to maximize the likelihood of observed data, employing a Negative Log-Likelihood for a Single Response (NLLsr) loss function, which was also applied in the subsequent conceptual models.

Further details regarding this neural network, including the precise calculation method for the distribution of responses and the

justification for its chosen configuration, are available in Supplementary Methods 3.

### Neural network as the guidance for model development

Despite their inherent interpretability challenges, it has been recently demonstrated that neural networks can provide valuable insights for theorists attempting to understand mental processes. In brief, the "scientific regret minimization" method utilizes the predictions produced by the neural network as guidance for the development of conceptual models[39]. This role of guidance manifests in two ways.

First, the predictions of an under-development conceptual model are compared with those of the guidance neural network to gain insights into what is missing in the former. Why compare against the predictions of the neural network rather than the actual data? The sophistication of neural networks allows them to approximate underlying mechanisms, effectively extracting genuine information within the data. Concurrently, regularization techniques help filter out the majority of the noise. As a result, the neural network's predictions may prove more helpful than the raw data when used as guidance for conceptual model development. Nevertheless, it should be noted that this comparison is made only for obtaining insights, and the model is always still fitted to the actual data.

Second, the predictions of the guidance neural network are used as virtual data, allowing us to go beyond the 10,000 patterns for which we actually have data and explore all $360^4 = 16,796,160,000$ possible patterns. Please see Supplementary Methods 5.3 for further details.

### Factorial comparison analysis as baseline for model development

Previous experimental studies of visual working memory have typically examined only one factor at a time. However, one study[17] conducted a factorial comparison of three factors—namely, the variability of mnemonic precision, the number of remembered items, and spatial binding errors—by simultaneously testing all 32 possible combinations of models (4 × 4 × 2). This factorial comparison study has greatly helped to clarify the issues surrounding the debate between slot[15] and resource[16]. It has become a landmark in this field.

Some of the levels within these three factors were inapplicable to the present design, resulting in a total of 8 possible models (2 × 2 × 2). Consistent with previous findings[17], the VP-F-NT model (Variable Precision, Fixed Capacity, with Non-Target Responses) emerged as the best among these 8 and is used as the baseline model for the present study. Further details of this factorial comparison analysis can be found in Supplementary Methods 4.

### A comprehensive exploration model

A quasi-comprehensive exploration model of VWM (QCE-VWM) is created to provide a comprehensive framework for VWM mechanisms, with "quasi" indicating a recognition of potential incompleteness. The QCE-VWM, similar to previous VWM models[15–17], uses explicitly interpretable mechanisms to simulate VWM's underlying functions but provides a more accurate fit to the data.

As depicted in Fig. 1c, the QCE-VWM model underwent iterative refinement (See Supplementary Methods 5.4). This refinement process —entailing the identification of mechanisms and the determination of their assembly—was guided by observational clues derived from the scientific regret minimization method and theoretical insights from existing literature, as listed in Table 1.

The structure of this model is shown in Fig. 3, which will be elaborated below. The relative importance of different mechanisms and aspects within the QCE-VWM model is shown in Fig. 4.

Similar to the guidance neural network, the QCE-VWM model aims to simulate observers' responses by mixing normally distributed components, representing knowledge-based responses, with a proportion of random guesses. Specifically, the model includes eight color-

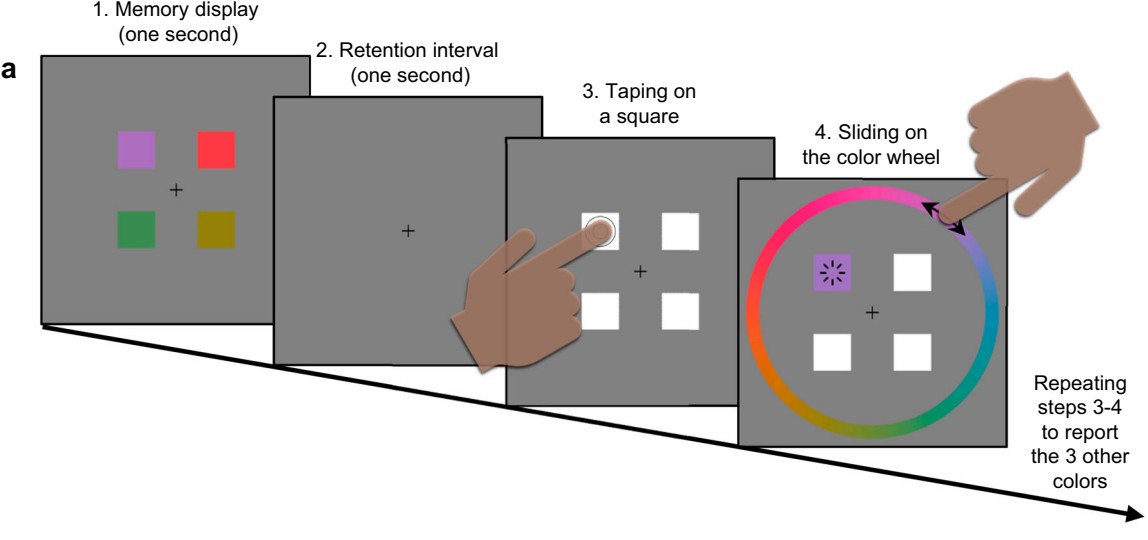

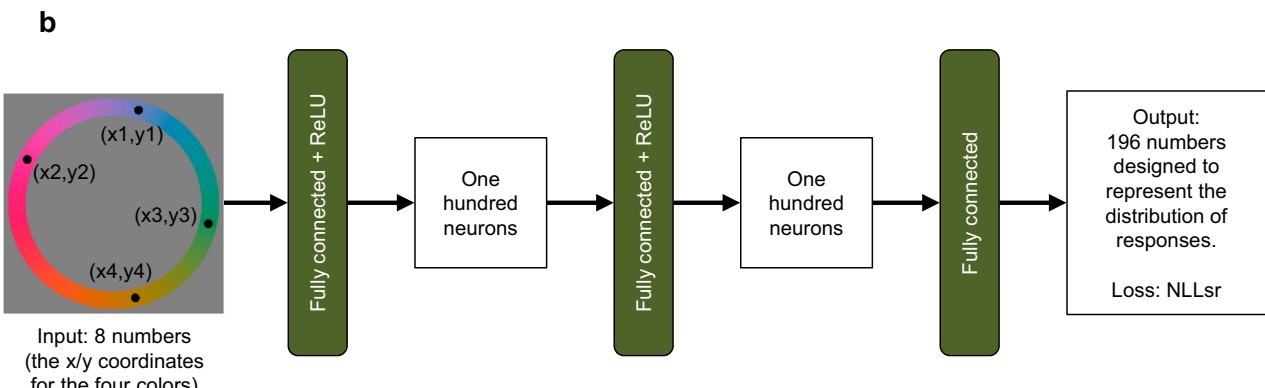

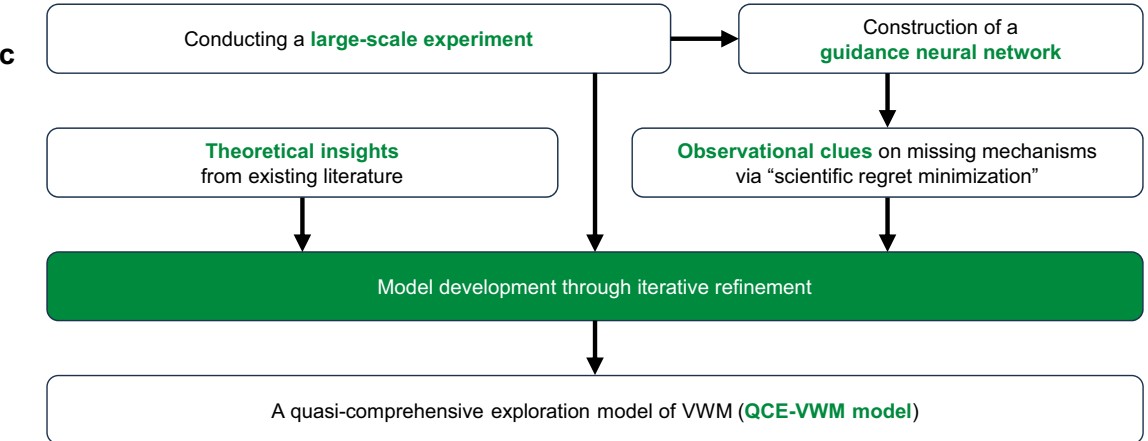

**Fig. 1 | Basic information of the present study. a** In the present study, a delayed estimation task was adopted. Participants attempted to memorize four saturated colors and report them after a 1-s retention interval. The report for each color was done by tapping the corresponding white square and selecting the appropriate color on a color wheel. **b** The guidance neural network includes four layers, respectively with 8, 100, 100, and 196 nodes and fully connected throughout. **c** The QCE-VWM model underwent iterative refinement. This refinement process, involving the identification of individual mechanisms and the determination of their interrelationships, was guided by both observational clues and theoretical insights.

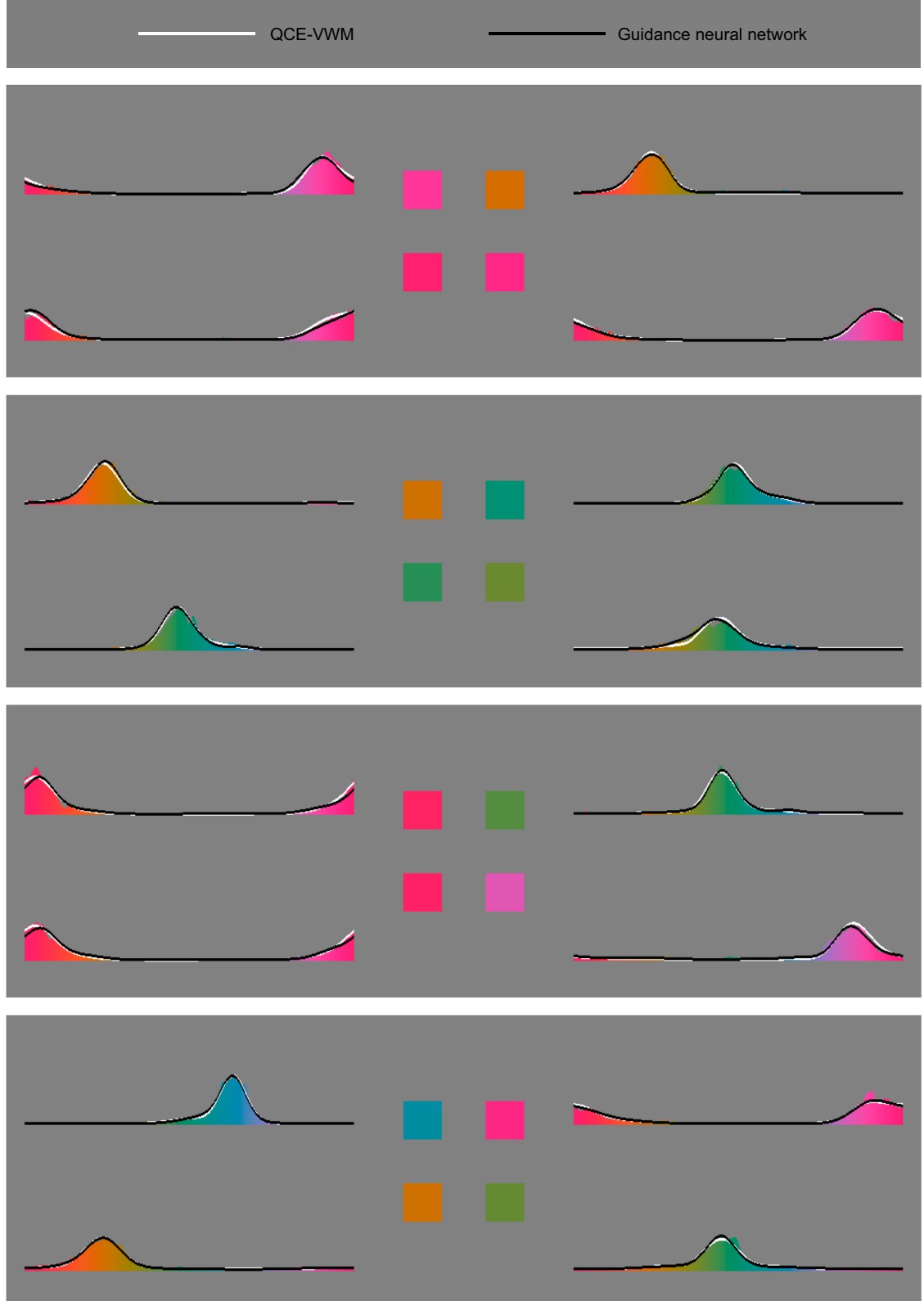

**Fig. 2 | Four sample patterns.** This figure shows the response distributions of four selected sample patterns, smoothed for clearer visualization. It also includes the predictions made by the QCE-VWM model (white curve) and the guidance neural network (black curve), both of which will be explained later. These patterns were selected due to their mid-range positioning in a comparison of the fitting accuracies between the two models, giving a fair visual assessment of their relative effectiveness.

**Table 1 | Findings and previous studies**

| Relevant mechanism | Implications | Previous studies | See also |
|---|---|---|---|
| Global-level implication | An integrative framework, created by merging individual mechanisms, is very effective, even more so than neural networks. | not previously reported | |
| Interactions between items | Items influence each other's biases and retention rates. | different from them[18] | 3.9 |
| | The interactions between items are based on pre-categorical, not category-based, color information. | not previously reported | 3.7 |
| | The effect of interaction on biases is governed by a Mexican-hat-like function. | confirms them[30] | |
| Chunking | Chunking magnitude = reduction in the number of storage units. Likelihood = between-chunk variability - within-chunk variability. | different from them[19,21,26,33] | 3.9 |
| | Chunking is based on pre-categorical, not category-based, color information. | not previously reported | 3.7 |
| | Better-chunked patterns are less likely to be swapped, and less attracted toward category centers. | not previously reported | |
| | Contrary to previous findings, better-chunked patterns are no more likely to be remembered. | different from them[19,21,26,33] | |
| Category-biased component | Memory of colors are affected by color categories. | confirms them[23] | 3.4 |
| | The category-based encoding is Bayesian-like but does not strictly follow Bayesian rules. | different from them[18,23,43] | 3.6 |
| | Red advantage: reddish colors are represented more precisely than other colors in category-biased component (i.e., red 2 category). | not previously reported | 3.5 |
| Unbiased component | Red disadvantage: reddish colors are represented less precisely than other colors in the unbiased component. | not previously reported | 3.5 |
| Swap-based component | Spatial binding errors occur at the representation stage, but not at the response stage. | different from them[13,20] | 3.8 |
| Concentration and Crosstalk | The weights of items affect each other. | not previously reported | 3.9 |
| Retention rates/ Random guess | More typical colors, as defined by the categories, are more likely to be remembered. | confirms them[27] | |
| | Consistent with the spirit of the slot model, only the retention rates, not the precision, are affected by interactions between items and spatial attention. | different from them[15,24] | 3.3 |
| Trade-off | There is a trade-off between the quantity and quality of representations. This is consistent with the spirit of resource model. | confirms them[16,28,41] | 3.3 |
| Precision of representations | There are low-precision components. | confirms them[17,29,31,32] | 3.2 |
| Spatial attention | Better-attended items are more likely to be remembered. | confirms them[15,51–54] | 3.1 |
| | Better-attended items' color categories are narrower and taller, and less effective at attracting the color-category biased component. | not previously reported | 3.1 |
| Category/ representation | The truncated normal distribution is superior to the von Mises distribution. | different from them[15,16,38,55] | 3.10 |

This table presents the primary finding of the QCE-VWM model, along with 20 specific implications. The "Previous studies" column provides references to relevant prior findings when available and clarifies whether the present study confirms them or supports a different conclusion. The rightmost "see also" column guides readers to the corresponding sections in the Supplementary Discussion where each topic is elaborated.

category-biased components in which the memorized colors are biased toward the centers of eight color categories, one unbiased component, and three swap-based components in which the memorized colors are replaced by the three other items of the four-color pattern.

The QCE-VWM model is divided into three distinct phases, each further subdivided into multiple steps. For a detailed, step-by-step explanation of the model, please refer to Supplementary Discussion 1.

## Phase 1: pre-categorical processing

Phase 1 incorporates two crucial processes that occur before color categories come into play.

In step 1a, interactions between items have two effects: an effect on the retention rates of items, which affects step 3d below, and an effect on bias, which affects step 3b below. The latter also influences multiple other steps by incorporating this bias into their calculations of color values. These effects are respectively represented by the blue and green arrows in Fig. 3. As illustrated in Figs. 5a, b, the influence of one item on another is described by a normal function of the color difference between the two items in the effect on retention, and by a Mexican-hat-like function in the effect on bias.

In step 1b, chunking between items is modeled. This chunking effect is represented by the red arrow in Fig. 3, and it affects two subsequent steps. As shown in Fig. 6a: the overall chunking effect of a pattern is calculated as a weighted average. On one hand, the elements being averaged are the chunking effects for all possible chunking

structures (see Supplementary Table 2). As depicted in Fig. 6b, the magnitude of the chunking effect for each structure is quantified as a reduction in the number of storage units. For instance, a two-chunk structure (e.g., 3 + 1) earns two points because it reduces the number of storage units from four to two. On the other hand, the weights used for averaging are determined by the likelihood of each structure, which is derived from the difference between between-chunk variability and within-chunk variability. Intuitively, the greater the between-chunk variability (items of a chunk are very different from those in other chunks), and the smaller the within-chunk variability (items within the same chunk are similar), the more likely a chunking structure is.

An important distinction between interactions between items and chunking is that the former focuses on the effects of interactions at the individual-item level, while the latter is a whole-pattern-level index that describes how well-chunked the entire pattern is.

## Phase 2: Calculation of weights of components

Eight color categories are defined in step 2a, each adhering to a normal distribution (see Fig. 7a). Importantly, there are two identical categories for reddish colors: red and red 2, a point that will be revisited below. In step 2b, weights of the eight color-category-biased components are determined by the distribution values of their corresponding categories. Intuitively, for blueish colors, the blue-category-biased component weighs more than the green-category-biased one, and vice versa for greenish colors.

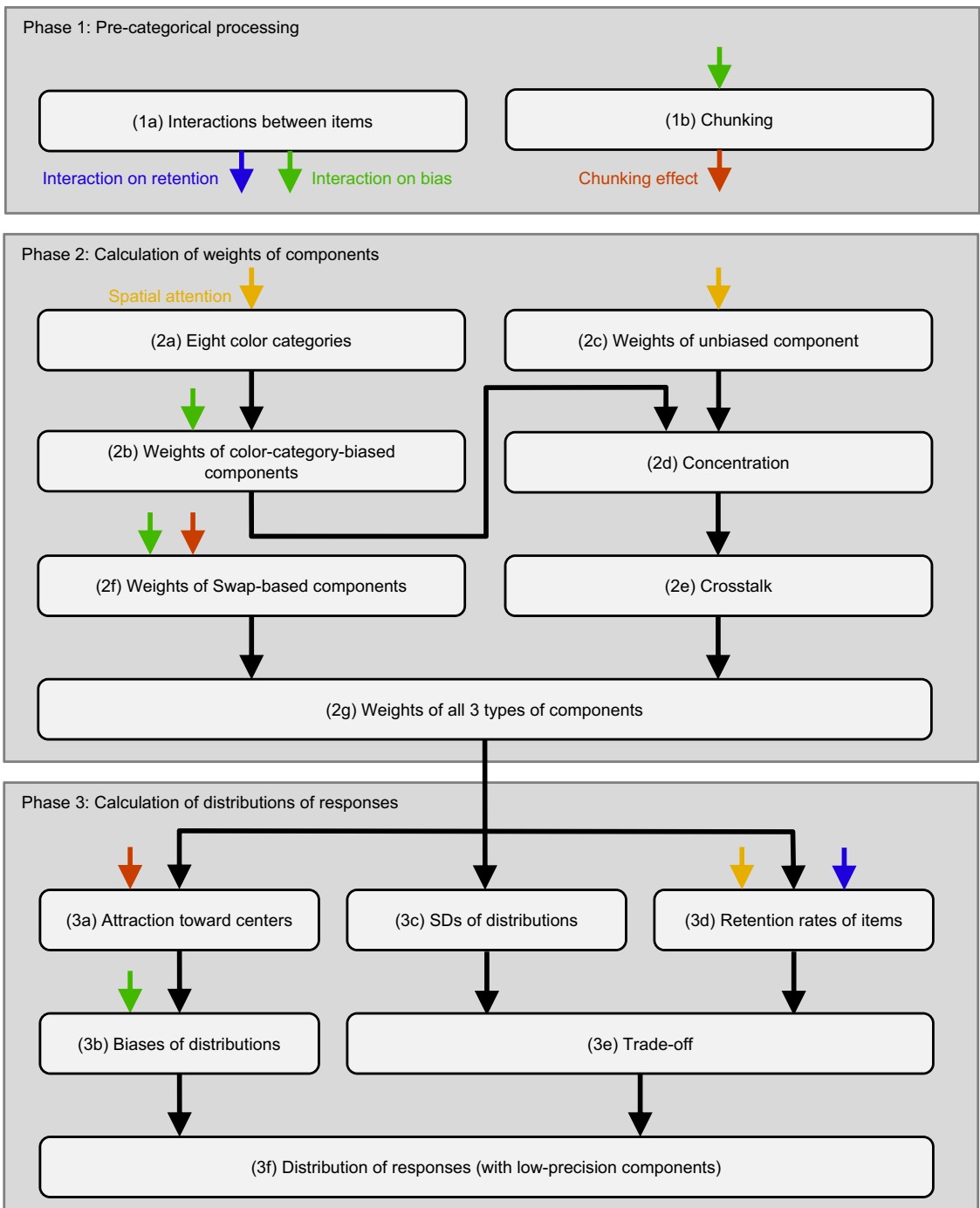

**Fig. 3 | Structure of QCE-VWM model.** This model aims to simulate observers' responses by blending knowledge-based responses with a proportion of random guesses. The former comprises three types of normally distributed components: eight color-category-biased components, one unbiased component, and three swap-based components. Throughout the operation of the model, the weights of these components, along with the means and SDs of their distributions, as well as the proportion of random guesses, are calculated and used collectively to make predictions about the distribution of responses. Blue, green, red, and yellow arrows respectively represent the effects of interaction on retention, interaction on bias, chunking, and spatial attention.

In step 2c, the weight of the unbiased component is determined by an item's location. This will be discussed below.

These nine weights—corresponding to the eight color-category-biased components and the unbiased component—then undergo two processes: concentration (step 2d) and crosstalk (step 2e). Previous VWM studies have explored the role of color categories[23] and the interactions between items[18] but have not examined the conjunction of these two concepts: how the categories of one item influence those of another. This study explores this conjunction, leading to the discovery of two mechanisms that have not been previously reported: concentration and crosstalk. In the concentration mechanism, smaller category weights are disproportionately reduced, which intensifies their diminishment—hence the term "concentration." By contrast, crosstalk involves a proportional redistribution of category weights between items, regardless of their initial magnitudes. Both concentration and crosstalk are influenced by the color difference between the two items, as illustrated in Figs. 5c, d.

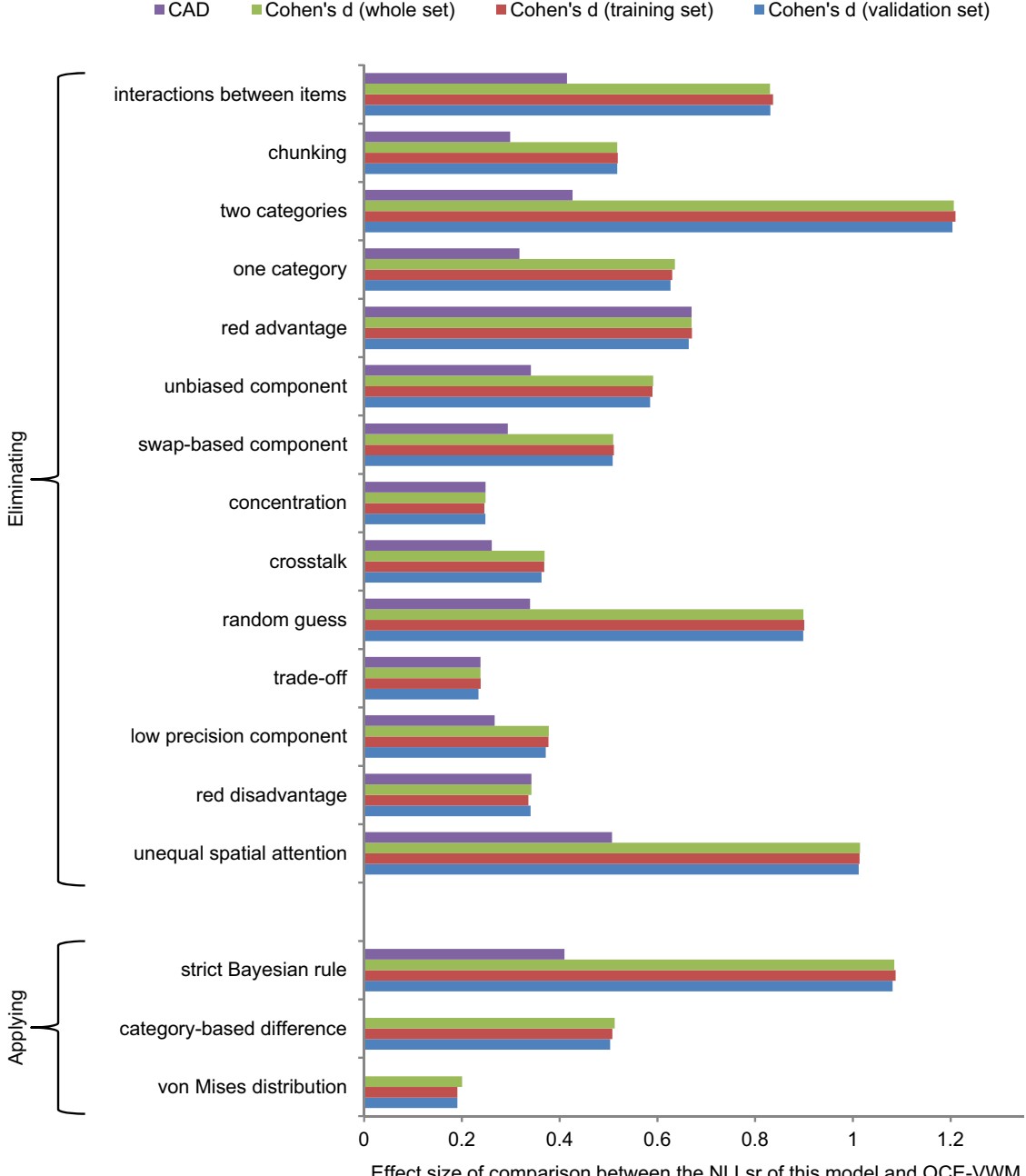

**Fig. 4 | QCE-VWM vs 17 alternative models.** This graph illustrates the cost associated with eliminating a single mechanism (or aspect) of the QCE-VWM model, or that of applying different methods (see text and Supplementary Discussion 2 for details). These costs are gauged by the effect sizes of t-tests in 17 comparisons between QCE-VWM and 17 alternative models. These effect sizes are measured as Cohen's $d$ value (green bars), as well as the CAD values (complexity-adjusted $d$, see Supplementary Methods 5.1, purple bars), and they are generally quite large. Moreover, cross-validation is performed, and the Cohen's $d$ value for the validation set (blue bars) is nearly as large as that for the training set (red bars), suggesting that these effects are generalizable.

In step 2f, the weights of the three swap-based components decrease as the target item and the swapped item become more different, as illustrated in Fig. 5e. Put simply, swaps occur only between similar items. Moreover, swaps are less likely to occur with better-chunked patterns.

In step 2g, the weights of all three types of components are combined.

**Phase 3: Calculation of distributions of responses**

In step 3b, the biases for all 12 components of the 40,000 items are calculated. Each of the biases for the eight color-category-biased components is determined as a proportion of the color difference between the category center and the item. This proportion, termed the "degree of attraction", is calculated in step 3a. It varies across categories and decreases for better-chunked patterns. By definition, the bias for the unbiased component is zero, whereas the biases for the swap-based components are the color difference between the swapped item and the concerned item.

In step 3c, the standard deviations (SDs) of the components are calculated. As illustrated in Fig. 7b, the SDs of the color-category-biased components are proportional to, specifically 80.8% of, the SDs of the categories themselves. However, there is a distinctive red advantage: the color-category-biased component associated with the Red 2 category is much more precise (SD ratio = 0.452) than the

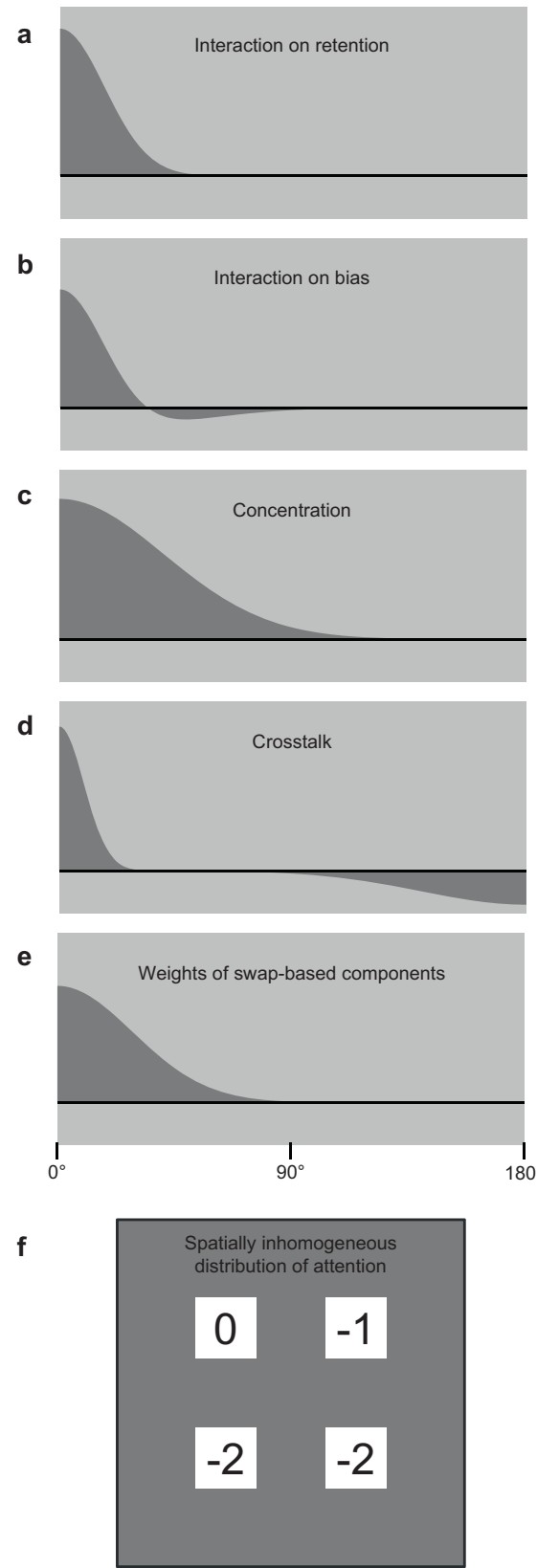

**Fig. 5 | Various mechanisms of the QCE-VWM model. a–e**. In five cases, a variable is the function of the color difference between two items. From top to bottom, the panels respectively show the functions governing the effect of interaction on retention rate, the effect of interaction on bias, concentration, crosstalk, and weights of swap-based components. **f** Several mechanisms have been identified to exhibit spatial inhomogeneity, as illustrated by the distribution shown here: the two bottom corners are at a clear disadvantage compared to the top-left corner, while the top-right corner occupies an intermediate position. This is likely attributable to the unequal distribution of spatial attention.

The SD of the swap-based components is constant, equivalent to 80.8% of the SD of the distribution of their weights (i.e., the distribution shown in Fig. 5e).

In step 3d, the retention rates of items are primarily calculated as the weighted average of the distributions of the eight categories on the color wheel, plus a constant baseline. This implies that atypical colors, falling between the primary color categories, are generally at a disadvantage[27]. The retention rates are also influenced by the interactions between items (i.e., the interaction on retention effect from step 1a).

In step 3e, a trade-off[28,41] takes place between the quantity (retention rates) and the quality (SDs) of VWM representations.

In step 3f, the response distributions are calculated. The response distribution for each of the 12 components of every item is derived from the previously calculated biases and SDs. These 12 components for each item are then integrated into a single distribution, which is combined with a low-precision counterpart (see also Supplementary Discussion 3.2) to form the overall distribution of knowledge-based responses–responses influenced by the knowledge about the colors of items. Then, the response distributions are computed as a mixture of knowledge-based responses and random guesses, with the proportion of knowledge-based responses indicated by the aforementioned retention rates.

## Spatial attention

The mechanisms of the QCE-VWM model are sometimes spatially inhomogeneous. As depicted in Fig. 5f, the top-left item has a distinct advantage over the bottom two items, while the top-right item occupies a middle ground. This phenomenon is likely influenced by reading habits[42], where readers typically begin at the top-left corner, proceed through the rest of the line, and then move to the lines below. In other words, this spatial inhomogeneity is probably a manifestation of the effect of unequal spatial attention to these locations and is tentatively interpreted as such (see also Supplementary Discussion 3.1).

As indicated by the yellow arrows in Fig. 3, this effect of unequal spatial attention impacts three steps. Specifically, better-attended items are more likely to be remembered (i.e., higher retention rates, step 3d), particularly as the unbiased component (step 2c). Its color categories are narrower and taller and are less effective at attracting the color-category biased component (step 2a).

## Statistical analysis

From a statistical perspective, the QCE-VWM model is robustly substantiated. It was compared with 17 alternative models. Fourteen of these models (models 2-15) were derived by eliminating a single mechanism or aspect, and they are used to show that each of these 14 mechanisms/aspects is essential for the QCE-VWM model. The remaining three (models 16-18) were developed by applying different methods to specific aspects of the QCE-VWM model, and they are used to show that each of these alternative methods is inferior to what is used in QCE-VWM.

Specifically, 17 t-tests were conducted to evaluate the QCE-VWM model's advantage (i.e., reduction in NLLsr values) over alternative

proportional relationship predicts. Interestingly, as shown in Fig. 7c, there is a red disadvantage in the unbiased component: reddish colors (at the category center) are much less precise (SD ratio = 1.827) than other colors. Altogether, reddish colors are unique[40] (see Supplementary Discussion 3.5).

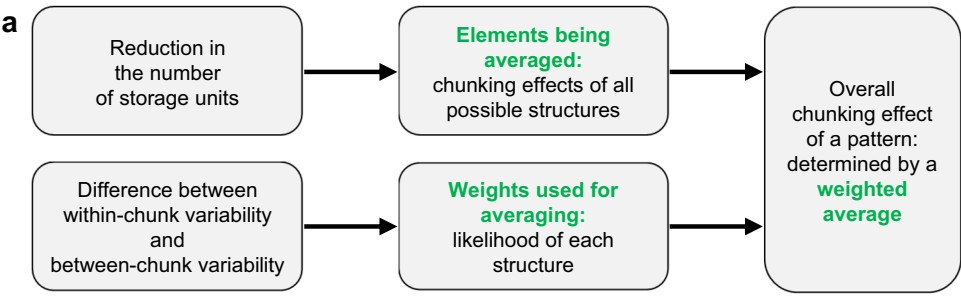

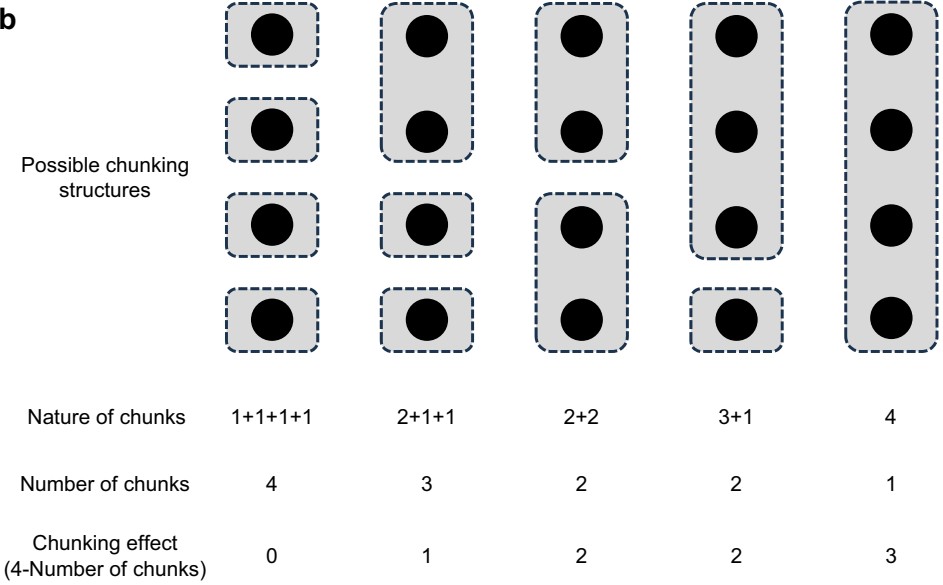

**Fig. 6 | Chunking. a** The chunking effect of a pattern is calculated as a weighted average across possible chunking structures. **b** The magnitude of the chunking effect for a structure is calculated as the reduction in the number of storage units.

models. Given the unusually large sample size (10,000 patterns), even small effects could produce extremely low p-values (p < 0.001 in all cases, see details in Supplementary Table 3). Therefore, p-values are not a suitable statistical index in this context. For an index that does not scale with sample size, the effect sizes (Cohen's *d*) of these 17 comparisons are illustrated in Fig. 4 (see the values in Supplementary Table 3). Most of these d values are decently large, but several are only slightly above 0.2, which could be considered small effects. However, Cohen's *d* is typically used in situations where the experiments are tailor-made to highlight one specific mechanism/factor, whereas the randomly-generated patterns in the present study are not. With this consideration, it seems fair to say that all these effects are decently large.

The mechanisms are not equally complex. For example, eliminating the chunking mechanism results in the reduction of 3 parameters, whereas eliminating the trade-off mechanism results in the reduction of only 1 parameter. Therefore, a "complexity-adjusted *d*" (CAD) is also presented in Fig. 4. This CAD is defined as follows (see Supplementary Methods 5.1 for further details).

$$CAD \text{ (complexity adjusted d)} = \frac{Cohen's\,d}{\sqrt{\Delta_{param}}} \qquad (1)$$

Furthermore, cross-validation was conducted to evaluate the generalizability of these 17 comparisons. The data were partitioned into 10 subsets, with models being trained on one subset and then applied to the other nine subsets. Figure 4 reveals that the Cohen's *d* values for these 17 comparisons are approximately the same for both the training and validation sets. The average generalizability ratio (Cohen's *d* for validation set/Cohen's *d* for training set) for the 17 comparisons is 99.5 %, suggesting that they are generalizable (see details in Supplementary Table 4).

## Discussion

Compared to traditional one-at-a-time studies, which yield fragmented insights like pieces of a puzzle, the comprehensive exploration approach merges these pieces of insight into an overall picture: an integrative framework. The task of putting puzzle pieces together is supposedly undertaken by literature reviews, but unfortunately, they are not very effective in doing so[9–11]. From this perspective, comprehensive exploration can be seen as an enhanced literature review equipped with a structured methodology. As depicted in Fig. 8a, it aims to amalgamate the precision and evidence-based nature of experimental studies with the extensive, holistic scope of literature reviews.

By integrating the fragmented insights, the QCE-VWM model has achieved an optimal balance between effectiveness and parsimony.

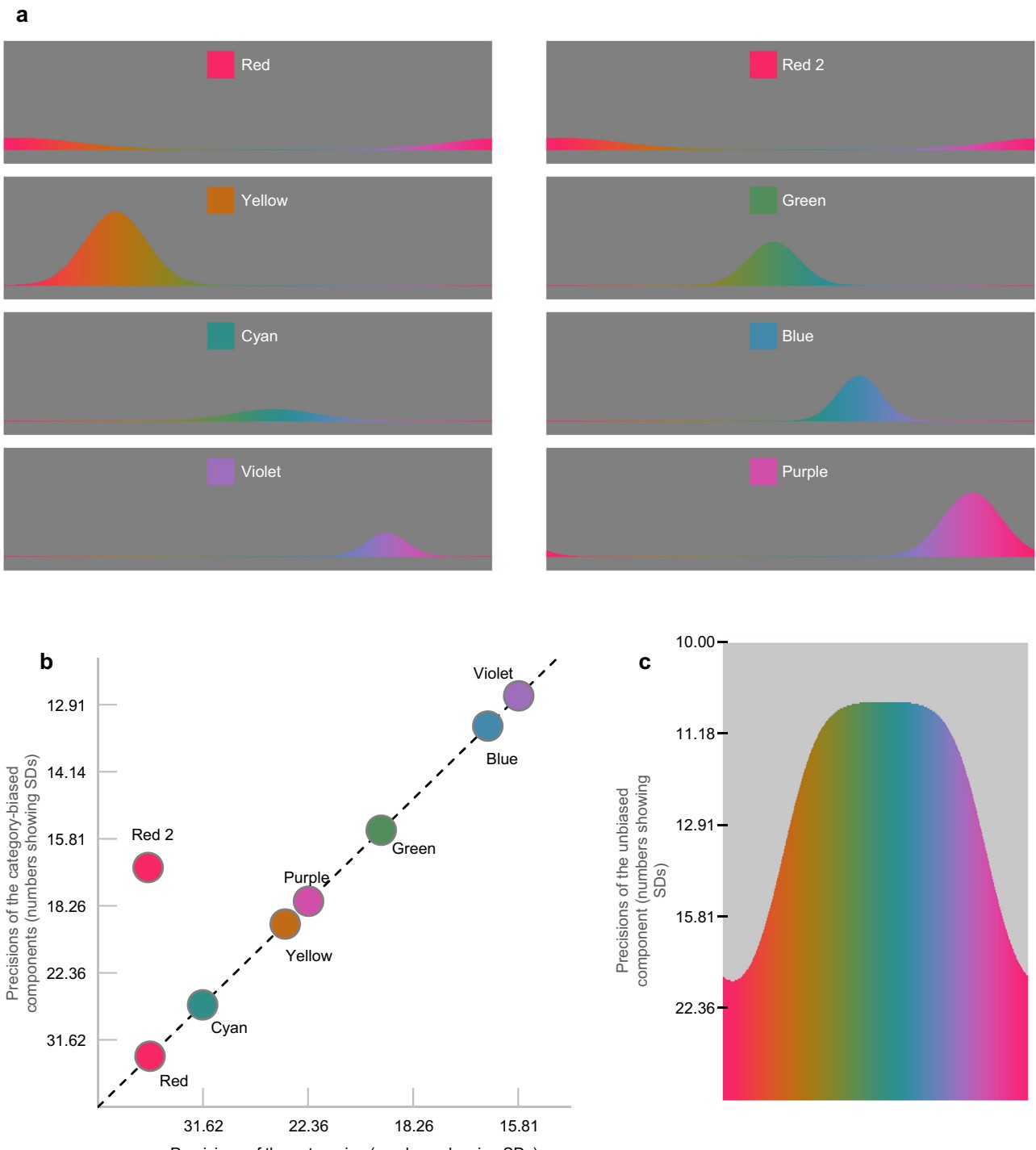

**Fig. 7 | Color categories. a** These eight panels illustrate the distributions of the eight categories on the color wheel. They determine the weights of the eight color-category-biased components. Note that two categories, Red and Red 2, share identical distributions but are included for the reason described below. **b** The SDs of seven color-category-biased components are proportional to the SDs of the categories themselves. However, there is a distinctive red advantage: the component associated with the Red 2 category is much more precise than the proportional relationship predicts. **c** Conversely, there is a red disadvantage in the unbiased component: reddish colors are much less precise than other colors. Please note that (**b**, **c**) use the scale of precision (i.e., $1/SD^2$), even though the numbers indicate the SDs.

Figure 8b compares the fitting of the QCE-VWM model with those of the guidance neural network and the baseline model (i.e., the VP-F-NT model discussed above). As expected, the guidance neural network exhibits a superior fit to the data compared to the baseline model. This aligns with the commonly observed discrepancy between neural networks and cognitive models: the former excel in data fitting but tend to be complex, while the latter are parsimonious but often fall short in effectively explaining data. The QCE-VWM model has achieved effectiveness but has also maintained relative parsimony. On one hand, it surpasses the guidance neural network in terms of data fit. The latter

is presumably a nearly full explanation of the data, so the QCE-VWM model also is. On the other hand, it remains fairly parsimonious, with 57 parameters compared to the neural network's 30,796. In brief, by summarizing the accumulated wisdom of decades of previous work on

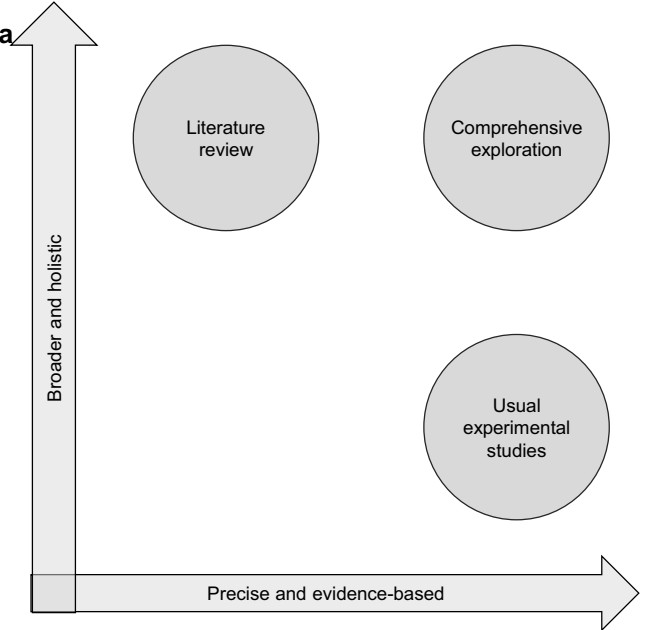

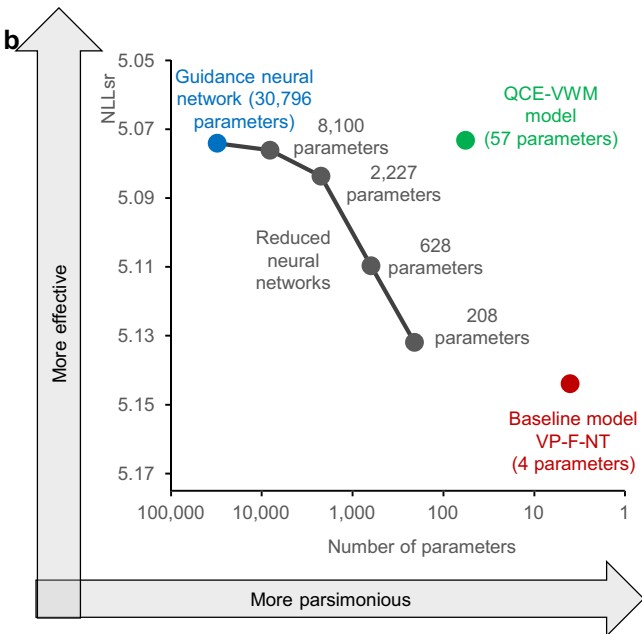

**Fig. 8 | Advantages of QCE-VWM model. a** The comprehensive exploration functions as an enhanced literature review, aiming to merge the precision and evidence-based characteristics of traditional experimental studies with the broad, holistic scope of literature reviews. **b** The guidance neural network and baseline VP-F-NT model have achieved effectiveness and parsimony, respectively. In contrast, the QCE-VWM model has managed to simultaneously achieve both; it surpasses the guidance neural network in terms of data fit and is also comparatively parsimonious. Four reduced versions of the guidance neural network are indicated by the gray dots. Their performance substantially decreases when the number of parameters is reduced to 628 and 208. This confirms that the neural network cannot remain effective without its complexity, highlighting the distinct advantage of the QCE-VWM model in simultaneously achieving effectiveness and parsimony.

VWM, the QCE-VWM model provides a better explanation of empirical observations than a massive neural network.

As shown in Fig. 8b, the performance of the neural network substantially decreases when the number of its parameters is reduced to 628 and 208 (see also Supplementary Methods 3). This observation further supports that the neural network cannot maintain its effectiveness without its complexity, highlighting the distinct advantage of the QCE-VWM model in achieving both effectiveness and parsimony simultaneously. For a complete comparison, all models involved in this study are presented in Fig. 9.

After demonstrating that the QCE-VWM model achieves an optimal balance between effectiveness and parsimony, we now turn to its conceptual implications. Using the puzzle analogy again, the key message of the present study is that a fairly complete overall picture is formed from these puzzle pieces, something rarely achieved or attempted for any cognitive task. In addition to this, there are numerous specific findings concerning the relationships among the puzzle pieces (i.e., the relationships between various mechanisms within the model), the discovery of new puzzle pieces, or the updating of old ones. There are too many to discuss exhaustively here; however, the 20 most important ones are listed in Table 1.

Next, we will go through five benefits of the comprehensive exploration over traditional one-at-a-time studies and/or literature reviews. First, a comprehensive exploration explicates the relationships among individual mechanisms. Traditional experimental studies examine different mechanisms one at a time and cannot elucidate the relationship among these mechanisms. For example, one previous study[23] distinguished between pre-categorical and category-based color information, while another explored the mechanism of chunking[21]. However, these separate studies do not address their relationship: is chunking based on pre-categorical or category-based color information? In contrast, a comprehensive formal computational model compels us to explicitly answer such questions. Specifically, by positioning chunking in phase 1, the QCE-VWM model implies that chunking relies on pre-categorical color differences rather than category-based ones (see Supplementary Discussion 3.7). To generalize, the QCE-VWM model automatically implies numerous relationships. For instance, are each of the aspects (e.g., retention rates, memory precision, categories' attraction, swapping) affected by each of the factors (e.g., chunking, interaction between items, spatial attention, and position on the color wheel)? The model provides answers to these and other potential questions, which can be found by examining the model's details (see Table 1).

Second, relevant to the preceding point, a comprehensive exploration facilitates the examination of the conjunction of existing findings. Previous studies have separately established the role of color categories[23], the interactions between items[18], and the role of random guesses[15]. The present study tries to explore the conjunctions between them. As mentioned above, the conjunction between category and interaction (i.e., how the weights of items affect each other) led to the discovery of two mechanisms that have not been previously reported: concentration and crosstalk (see Supplementary Methods 5.4 for the other conjunctions). Such a conjunction-based finding is unlikely to emerge from traditional one-at-a-time studies because it requires the simultaneous consideration of two factors that are not typically considered together. Nevertheless, their importance for explaining the data (Cohen's $d = 0.248$ and $0.261$, respectively, for concentration and crosstalk) are comparable to the conceptually straightforward mechanism, trade-off (Cohen's $d = 0.238$).

Third, in the comprehensive exploration, a missing or redundant mechanism can be objectively assessed through model fitting. On one hand, if an important mechanism is missing from the model, it will result in a set of poorly-fitted patterns, allowing us to speculate on the nature of the missing mechanism. On the other hand, if a mechanism is redundant, then it will not lead to further improvement in the model's

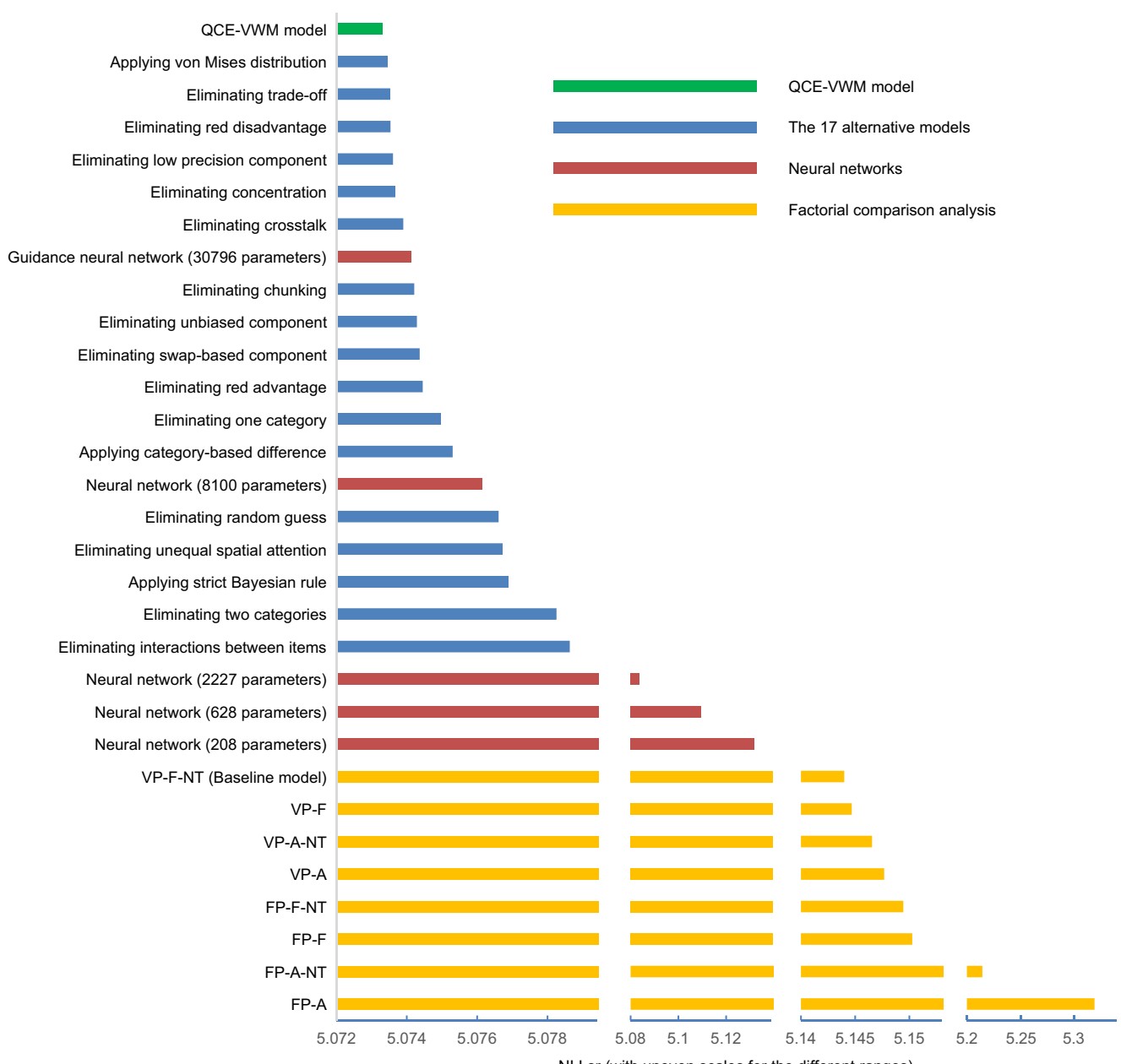

**Fig. 9 | A thorough comparison of all models.** The QCE-VWM model (represented by a green bar), along with the 17 alternative models (represented by blue bars), the guidance neural network and its four reduced versions (represented by red bars), and the eight models in the factorial comparison analysis (represented by yellow bars), are all sorted based on their data fitting (NLLsr). Please note that uneven scales are used across different ranges because the fittings are densely clustered within two specific ranges: one for the QCE-VWM and alternative models, and another for six of the eight models in the factorial comparison analysis. Consequently, uneven scales are employed to highlight the distinctions within these two clusters. The QCE-VWM and alternative models greatly outperform those in the factorial comparison analysis due to their effective integration of more mechanisms. Furthermore, the QCE-VWM and several alternative models surpass the performance of the guidance neural network, presumably because the latter does not capture the underlying mechanisms as precisely as the former.

fitting. In comparison, the literature review is inherently subjective and lacks an objective method to guarantee the detection of missing or redundant mechanisms.

Fourth, a comprehensive exploration fosters more constructive theoretical developments. Consider, for instance, the ongoing debate[15–17,20,24] between the slot model and the resource model, which has significantly influenced studies on VWM over the past 15 years. The slot model proposes a fixed number of slots for remembering items, storing high-precision information for items assigned a slot, and making random guesses for others. In contrast, the resource model suggests flexible resource allocation, enabling potentially unlimited items to be remembered with varying precision. As mentioned above, the factorial comparison study[17] has bridged some of the gaps between them. The QCE-VWM model further progresses in integrating these theoretical positions. On one hand, it asserts that only retention rates, not SDs, are influenced by external factors (spatial attention, interactions between items), making the SDs a more fixed aspect than the retention rates. This aligns with the spirits of the slot model. On the other hand, the trade-off observed between the quantity and quality of representations in Step 3e is consistent with the spirits of the resource model. Overall, it is evident that both the slot and resource models encapsulate certain aspects of the truth, yet neither is entirely accurate

(see Supplementary Discussion 3.3 for more details). The QCE-VWM model serves as a constructive intermediary in this debate, amalgamating and scrutinizing insights from both the slot and resource models within an integrative framework.

Fifth, a comprehensive exploration is more precise. For instance, the color-category-biased components are attracted toward the category centers. Although this center-attraction mechanism appears Bayesian-like, substantial modifications are needed. Bayesian rules predict an additive relationship between the precision of color-category-biased components and the precision of those categories[18,43], yet a multiplicative relationship provides a better fit to the data (see Supplementary Discussion 3.6). For another example, while the von Mises distribution is commonly considered the appropriate substitute for the normal distribution in circular space, the current analysis reveals that the truncated normal distribution offers a better explanation for the data (see Supplementary Discussion 3.10).

After reviewing the benefits of comprehensive exploration, we will now explore aspects that some may find undesirable, beginning with the issue of the model's complexity. One might argue that the QCE-VWM model, with its 57 parameters, is overly complex by cognitive psychology standards. However, the traditional belief that a model should be limited to a few parameters emerged in contexts with limited datasets. Applying this convention to large datasets can be misleading. Recent research indicates that as datasets expand, the optimal models should also become more intricate[7,44]. Furthermore, the mechanisms underlying any function of the human mind are likely to be multifaceted. Therefore, if the goal is to reveal this complex truth, then the model must be correspondingly complex. From another perspective, as illustrated in Fig. 8a, the comprehensive exploration can be considered an enhanced literature review. When the QCE-VWM model is compared to a recent literature review[35], it demonstrates a comparable breadth of factors considered. Therefore, the complexity of the QCE-VWM model aligns with the expectations for an enhanced literature review.

A consequence of complexity is the iterative nature of the model. For example, the factorial comparison analysis[17], which explored a solution space of $4 \times 4 \times 2 = 32$ possible models, is broader than what is typical in experimental psychology. The present study shares the goal of simultaneously testing multiple factors. However, the current QCE-VWM model has 57 parameters, making it obviously impossible to exhaustively test all $2^{57} = 1.4E + 17$ ablated models, not to mention the many other parameters that could have been included. Therefore, unlike the factorial comparison study, which conducted exhaustive testing within a predefined space of solutions, the present study adopts the style of AI studies: a data-driven iterative search within an unlimited space of solutions. Thus, the current QCE-VWM model, while being the best option available at this moment, is tentative and may be replaced by superior alternatives in subsequent iterations. Several points need clarification regarding this iterative nature.

First, the minor cost of being iterative is outweighed by the greater benefit of exploring unlimited possibilities. This is why the QCE-VWM can surpass factorial comparison analysis[17]. While exhaustive testing is valuable because it can identify the best solution within a predefined space, isn't it preferable to discover an even better solution by venturing into a larger space?

Second, being iterative does not equate to being arbitrary. Although the existence of essentially unlimited possible models prevents exhaustive assessment, necessitating an iterative search, those models that are assessed undergo rigorous statistical evaluation. The statistical evidence provided underscores the indispensability of all the model's mechanisms. Additionally, many candidate mechanisms, including the Boolean map[45,46], have been tested and found to be unhelpful and thus were rejected, indicating that mechanisms cannot be arbitrarily added. The Boolean map is particularly worth mentioning because it was the primary driving force behind my theory-driven studies[45–48] for 15 years and indeed the initial reason I started this project. However, it had to be rejected because the data indicated as much. This example clearly demonstrates that there is little room for subjective bias in the decision to add or remove a mechanism.

Lastly, the iterative or tentative nature aligns well with the inherent process of scientific discovery[49]. In the AI domain, the iterative approach to model development is often viewed as a strength because it enables starting with a modest proposal, gathering feedback and additional data, and then refining the model based on those inputs, continuing this constructive loop of enhancements. This principle certainly applies here. Beyond the QCE-VWM model, the current dataset also represents an initial attempt. Hopefully, this study will inspire peer researchers to adopt a more comprehensive approach, and ultimately, the community will decide how to establish a better benchmark dataset for everyone's use.

## Methods

The experimental procedure adhered to The Chinese University of Hong Kong's guidelines for conducting survey and behavioral research. Ethical approval was obtained from the Research Ethics Committee of The Chinese University of Hong Kong prior to the commencement of the study (SBRE-19-224, approved on 6 February 2020; SBRE-21-0204, approved on 6 December 2021). This approval encompassed the consent form, the experimental processes, and the payment system involved in the experiment.

The experiment was conducted as an online game that participants accessed using their personal devices. For obtaining informed consent, participants were explicitly informed that the outcomes of the testing would contribute to a scientific study led by the author. Additionally, details regarding the task embedded within this experiment were shared with them. Participants expressed their willingness to take part in the study by tapping the Continue button on their personal devices.

### Online data collection platform

The current experiment was conducted as an online game using our laboratory's online data collection platform (https://huang.psy.cuhk.edu.hk/games/). The computer code used to create the webpage for data collection was written in JavaScript, Vue.js (version 2.6.11), and PHP (version 5.3.3).

This platform is embedded within the WeChat app, meaning the webpage can only function properly when accessed through the app. This integration with WeChat is essential for facilitating user engagement and management.

WeChat, a widely used multipurpose instant messaging and social media application among Chinese individuals, boasts over a billion active users. This makes it convenient for users to share the platform on their WeChat Moments. More importantly, embedding the platform within WeChat enables access to its identification system, streamlining user payment processes and preventing the creation of multiple IDs. To utilize WeChat's ID system, the platform must automatically detect WeChat IDs, which necessitates opening the webpage exclusively within the WeChat app.

### Participants

As described, the experiment was conducted as an online game that participants accessed using their personal devices, indicating that the majority were likely active internet users. Additionally, the data collection platform was presented in Chinese and embedded within the WeChat app, implying that participants were probably Chinese language users. Individuals with color vision deficiencies were explicitly instructed not to participate. Beyond these factors, no other apparent bias exists in the study population.

A total of 2316 participants (59.3% female; mean age = 29.4 years, both based on self-reported gender and age) participated in the game.

During each week-long session, several hundred active participants received participation-based awards (25 Chinese Yuan each), while two randomly selected participants were granted lottery-based awards (500 Chinese Yuan each). Further details can be found in the Supplementary Methods 1.3.

The target dataset size was determined based on an estimation of the total number of trials required. Specifically, based on previous experience with working memory studies, it was estimated that 1000 trials would be sufficient for measuring VWM for each individual color pattern. Consequently, a total of 10 million trials was planned.

### Stimuli and procedure of the experiment

The current working memory task was depicted in Fig. 1a. Each trial began with a 1-s presentation of a fixation, followed by a memory display that showed four colored squares arranged in a 2 × 2 matrix. Participants were instructed to memorize these four colors. This memory display lasted for 1-s, succeeded by a retention interval of equal duration. Upon completion of the retention interval, four white squares appeared, indicating to the participants that they could initiate their response.

Participants responded by tapping one of the white squares and then sliding on a color wheel to report the memorized color of the selected square. The orientation of the color wheel was randomized for each response. During sliding, the color of the selected square immediately changed to reflect the currently selected color. Once satisfied with their choice, participants released their finger to confirm their response. They then repeated this process for the remaining three colors. In this procedure, colors that had already been reported remained visible. Participants responded to the four items in any order they preferred.

The aforementioned displays occupied the central square area of the device. On a typical mobile phone screen, the squares and the gaps between them measure 0.91 cm and 1.04 cm, respectively.

Randomization was utilized for both the generation and use of color patterns. In each trial, the colors were randomly selected from a pool of 10,000 color patterns. These patterns were generated in advance, each by selecting four random integers from the range of [0, 359]. Each number corresponds to a color, represented as an angle on a color wheel. Since participants used their own devices, the actual colors displayed inevitably varied slightly between devices, implying slight inconsistencies from their designed values in the color space (See Supplementary Methods 1.2 for further details).

### Preliminary analysis and data exclusion

The dataset incorporates a total of 10,159,250 trials, encompassing 40,637,000 responses. The mean performance was commendably good, mirroring the results of past studies. Specifically, in preliminary analysis, performance is measured by the average root mean square error (RMSE) of participants' responses in comparison to the actual colors on the color wheel. The average RMSE for the dataset is 47°, suggesting that participants generally focused on the task at hand.

Certain trials were excluded based on pre-set criteria. Poor blocks, characterized by an average RMSE of 90° or more, were excluded. This led to the elimination of 0.64% of trials. Unusually good blocks, characterized by an average RMSE of 10° or less, were also eliminated. This resulted in the removal of 0.025% of trials. This second exclusion was motivated by the assumption that such outcomes are likely the result of artificial strategies. Following these exclusions, the dataset contained a total of 10,091,320 trials and 40,365,280 responses for subsequent analysis.

Next, the response distributions for the 40,000 combinations (10,000 patterns × 4 colors) were computed by amalgamating the responses from all trials featuring the same pattern. The compiled data was then utilized by the neural networks and the QCE-VWM model for their respective modeling processes, implemented using PyTorch version 1.12.0 and MATLAB (R2022b), respectively.

### Reporting summary

Further information on research design is available in the Nature Portfolio Reporting Summary linked to this article.

## Data availability

All data is available on the Open Science Framework[50] and can be accessed at https://doi.org/10.17605/OSF.IO/QPY49.

## Code availability

All scripts used for data analysis are available on the Open Science Framework[50] and can be accessed at https://doi.org/10.17605/OSF.IO/QPY49.

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

## Acknowledgements

The work described in this paper was supported by the Research Grants Council of Hong Kong (CUHK 14610520 & CUHK 14606622, both awarded to L.H.). The funder had no role in study design, data collection and analysis, decision to publish or preparation of the manuscript. The ChatGPT has been utilized to enhance grammar, spelling, and phrasing of this paper, but it has not been employed to generate any new text.

## Author contributions

L. H. is the sole author of this article.

## Competing interests

The author declares no competing interests.
