## [Transparent Peer Review file · Nature Communications]

Comprehensive Exploration of Visual Working Memory Mechanisms Using Large-Scale Behavioral Experiment

Corresponding Author: Professor Liqiang Huang

Version 0:

Reviewer comments:

Reviewer #1

(Remarks to the Author)

This paper uses a massive data set (10 million trials and 40 million responses) to fit and test models. I think this approach is very intriguing and I also value the development of theoretical models as is done here. The author provides a comprehensive guide to how to replicate these analyses, although I did not replicate them myself. All of the files and data are available on OSF which is also commendable.

My biggest concern with this paper is that it is not clear what I have learned at the end of reading it. As there is no clear cut theoretical outcome. It is clear that a massive neural network does an overall better job of fitting the results in terms of absolute fit, but the argument is that the 74 parameter model is a better bargain because of the reduced complexity. That's all well and good, but what am I, a cognitive neuroscientist supposed to take from this finding that is going to affect my understanding of working memory?

Another concern with the results is that the relative difference between the preferred model and the alternative/simplified models might be extremely small, and the statistical significance observed is a product of the large amount of data. However as a reader it is hard for me to understand the significance of the results, both in terms of statistical significance, and the magnitude of the effect. The critical issue is whether these specific qualitative features are important, because there are so many different parameters and in the QCE-VWM model. It is possible that with a model this complex, the actual components aren't that important, and nearly any combination of encoding mechanisms could be adjusted to match human performance similarly well. The author attempts to address with the alternative model analysis but I was unable to understand what these analyses really implied.

To explain, with data sets this large, it is hard to have a good intuition about what an AIC of e.g. 20,000 means, since AIC scales with the number of data points. 20,000 in one experiment might indicate an enormous loss in fit because the # of data points is fairly low, while in another experiment with more data, like this one, a tiny loss in the quality of the fit might produce the same massive AIC difference. A better approach would seem to be something that quantifies how much fit is lost with a measure that does not scale with the number of data points, such as % variance explained, or root mean squared error. You could then boot strap distributions of RMSEs for multiple fits of intact vs degraded models. The paper also uses NLLsr, but I am similarly unable to intuit what this means. For example, what does a drop from 5.08 to 5.075 imply about the quality of fit? This is a paper that I presume is targetted at the visual cognition community and its results should be presented in a way that are relatively straightforward to understand if it is to have an impact.

It is strange that order of report is not analyzed. It is well known that earlier reports have higher accuracy than later reports, and this is especially true when participants are allowed to select the order for report. This seems like a critical variable because the retrieval of the first vs the 4th color may differ in terms of what qualitative features are present. c.f. Adam, K. C., Vogel, E. K., & Awh, E. (2017). Clear evidence for item limits in visual working memory. *Cognitive psychology*, 97, 79-97.

Fig 8b is misleading as the scale suggests that the difference between 2 and 74 parameters is pretty small, but this isn't true. The difference between 2 and 74 parameter models is enormous in terms of flexibility. This should perhaps be a log scale to allow us to see the difference between the smaller numbers

I suspect you could trim down the neural network substantially and keep a similar fit, this analysis should be attempted. For

example, does a greatly reduced parameter version of a neural network achieve similar performance as the 74 parameters QCE, maybe 200 parameters or so? It would take some engineering to get the number of parameters that small given the number of inputs and outputs, but I suspect you could get the # output nodes down to 8 also, and

Also I was not clear why the sin/cosine of the colors were used as inputs. The sin/cosine method of input doubles the number of input nodes and perhaps only adds confusion to the model. For example, the cosine is equal at 90 and 270 degrees but those colors are not more similar than the colors at 90 and 180 degrees are. So the input transform creates an inferential bias in color similarity that the model has to learn to overcome. A better solution might be to put the colors on a grid and use the x/y coordinates of the color on the grid. This would ensure that colors close to each other in both x and y are more similar colors. However maybe this doesn't greatly affect the results but I had a hard time understanding the rationale.

I could not understand the generalization rate analysis of S8.4. What does 100.5% mean and how was it computed?

On a technical note, in the AIC comparison, if one of the features of the model is set to 0, how many parameters does that remove from the total, and is that change reflected in the AIC?

In the Preliminary analysis section: "with an average Root Mean Square Error (RMSE) of 47° on the color wheel." I could not understand how the error was being computed. Was this the error of participant responses for all four responses relative to their actual colors?

I could not understand exactly what the dashed line represented in supplemental Figure 1.

Reviewer #2

(Remarks to the Author)

In this manuscript, the author reports the results of a massive experiment to understanding visual working memory (VWM), building upon the author's prior work in applying the scientific regret minimization technique to visuospatial memory. After collecting a new benchmark dataset of 40 million behavioral responses over 10,000 color patterns, the author develops a new model of VWM, termed QCE-VWM. The model, which is simultaneously less complex (and more interpretable) than traditional neural network models while avoiding the often squishy verbal nature of model specification in this domain, surpasses them in data fitting, showcasing the effectiveness of the scientific regret minimization method in this domain. By integrating concepts drawn from both slot and resource models, the study attempts to reconcile conflicting theories in VWM. It also introduces new mechanisms like the Matthew effect and crosstalk.

In general, I find the work promising and the dataset alone would be enough for the paper to be a valuable resource to the community. That being said, I have several major concerns about the work in its current form.

1. The Matthew effect and the crosstalk effect are each defined in only a single sentence in the main text, which is too little justification for what even the abstract claims as a contribution to the literature with respect to novel working memory mechanisms. The supplement goes into more detail about their respective definitions, but provides no motivation in terms of how these effects relate to what is known, nor how these effects were discovered.

2. The link between the proposed model's design space and those of extant models in the literature is insufficiently clear. I would have strongly preferred a model comparison like that in the van den Berg et al. (2014) factorial comparison paper, Fig. 3c, where the proposed model, all the ablated versions, as well as all the major extant models from the literature are compared head to head. My guess is that they will perform quite well for a narrow slice of the stimulus space, and quite poorly for random samples of the stimulus space. Taking this approach has the dual function of showing the community that the work has taken previous formal models in this domain seriously, while also showing the community that those previous models fail when considered at scale.

3. It is unclear what the inclusion/exclusion criteria on working memory mechanisms are that justify an 18-feature design space of models. Many of these model components are indeed well-supported by the literature, but there are other components that might have been included but were not (e.g., positional effects, kurtosis from cross-participant or cross-trial variability in precision, time-of-day effects, serial dependence, interference across trials in a run, interactions between iconic and working memory, display resolution, participant-level features like gender, etc.). The paper includes a bit of hedging to address this (e.g., if anything is missing, a new model with the added component could be tested and might surpass the present model), but that doesn't help address the specific question of what's in vs. out for the present work. Was some kind of systematic literature review completed?

- Apologies if I am being dense, but I did not catch what the QCE acronym in the model name expands to.

- The supplemental materials are thorough and do an excellent job of providing the information needed to replicate the work.

Reviewer #3

(Remarks to the Author)

This massively large-scale study employs a whole-report color visual-working-memory task and models the results with a

convolutional neural network to estimate how well the data can be explained by a flexible data-driven model. On each trial of this task, administered on participants' own devices (most likely mobile phones) via the multi-purpose personal app WeChat, participants saw four colors and subsequently had to report all four colors in a self-determined order on a color wheel. The main achievement of the current study is the development of a theoretically derived framework model that can account for the data as good or even better than the neural network. Based on the full model the necessity of its various theoretically derived component mechanisms is evaluated. It turns out that even mechanisms that were previously considered competitors for explaining the same data patterns all explain unique portions of the variance.

This is a highly interesting novel approach to cognitive psychology that I was not aware of before and I will definitely follow the author's work with this approach closely from now on. The sheer amount of data collected is impressive, although a tradeoff with regard to data quality had to be made (see in particular Point 1 below). The modeling takes a large number of known influences on VWM performance into account (chunking, spatial biases, color categories, attraction/repulsion, swapping, quantity/quality tradeoff, guessing...) and also considers additional influences (Matthew and Crosstalk effect). It is therefore also far more extensive than any previous modeling in this area. The exploration of model versions documented mainly in the supplement seems appropriate and theoretically well informed. I've read all materials with great interest and believe that this work will have a profound and persisting influence on my own thoughts on VWM. This work is useful for the large number of new specific and testable hypotheses alone, but it provides so much more. I also found the information on the author's webpage on the development of this approach highly interesting and recommend reading it as well: <https://sites.google.com/view/liqianghuang/story?authuser=0>

Signed. Heinrich Liesefeld

Here are my minor comments that hopefully help to improve the manuscript even further:

1. There should be a disclaimer in the supplement and paper that the colors were *not* CIE-L*a*b, because the screens were not calibrated; I assume that colors were L*a*b* under the unlikely assumption that screens were calibrated to sRGB. Also note that variation in colors between devices likely contributed to interindividual variance in performance and to complex distortions of the color space. Could this inaccuracy in color presentation account for the deviation from the Bayesian predictions (e.g., Suppl. Fig. 3)?

2. The author strikes a great balance between detail and brevity in his writing. I only found a few aspects that were not yet sufficiently explained for my taste:

2a. I think more detailed explanation of the "scientific regret minimization" approach is needed for the reader to understand this paper. It is partially given in the section "Neural network as the target for model development" (p. 4), but some more elaboration and clearer indication which of these ideas are at the core of scientific regret minimization and which have been added for the current purposes would be helpful, I feel. I was somewhat confused by the fact that the approach is introduced in these terms and then there is a statement that the model did "surpass the target neural network in terms of data fit" (p. 7) – how and in which way can it surpass the neural network model if the neural network model was the target? Some confusion needs to be resolved here. "the QCE-VWM has made considerable progress, outperforming the target neural network" (p. 10) indicates that I misunderstood what "target" means here.

2b. "ReLU activation function" is mentioned, but not explained.

2c. Given that the Matthew and Crosstalk effects are introduced to this community here, a more detailed explanation seems warranted on p. 6. This should contain a detailed explanation in how far these types of effects are not already covered by previously proposed VWM mechanisms.

3. I believe that readers would appreciate core references for each of the introduced VWM mechanisms in the supplement (as far as these have been proposed before). What I have in mind here is that for each model component there could be a reference to 2-3 core papers that have suggested, empirically shown, or reviewed the respective cognitive/neuronal mechanism. If the author starts collecting these references, I'm happy to supplement them based on my knowledge of the literature during the next round of reviews.

4. It is claimed that the conversion to cosine and sine values "indeed improves the predictions of the neural network." (Suppl., p. 3) – but compared to which alternative measure of color does it improve predictions? I can see that the apparent dissimilarity between, say, 258° and 2° on a linear scale would cause trouble for the model, but is there really no proper solution for handling circular spaces using just a single value/input neuron?

5. Instead of asking the reader "to use document comparison software to highlight the changes" (Suppl., p. 13), delta versions could be added to the OSF project.

Version 1:

Reviewer comments:

Reviewer #1

(Remarks to the Author)

I think this version of the paper is a marked improvement over the previous one but I still think it could be substantially

improved by some additional revisions.

On this second review I have now come to understand its value, which does not come out very clearly in its current form. Essentially this paper is a comparison of the effectiveness of the accumulated explanatory power of cognitive science to explain working memory against a theory-free neural network. The value of theory has always been ascribed as providing concise (i.e. low parameter) descriptions of natural phenomena (hence the phrase "to carve nature at its joints"), and the demonstration here is indeed that the accumulated wisdom of decades of work on WM does indeed provide a better explanation of empirical observations than a massive neural network, despite the difference in parameters. This is a valuable and reassuring demonstration of the effectiveness of good theory.

However this framing is somewhat obscured in the introduction which connects the QCE approach to Artificial Intelligence given that a large amount of data is used to parameterize the model. Just because an approach is driven by large data sets, that does not mean it is AI. One could even argue that the QCE approach here seems more like the opposite of AI as it is currently construed in the literature, given that it is using theory derived from experimental work to form the basis of the candidate model rather than just fitting a massive set of parameters (which is what the neural network is doing).

This is not my paper of course and these are opinions about framing, rather than observations of factual errors, so the author and editor may disregard.

Another issue is that the framing of the results section could be improved. I would start off by making it clear that there are three models that will be compared, and explain why these models are being used. In the current form the reader has to puzzle out that there are three different models and the terminology is confusing. e.g. the neural network is a guidance mode and the VP model is the baseline. Why these terms? I don't understand. To me, the neural network is a baseline of how good the fit can get without theory. The VP is an alternative theoretical model to QCE. VP and QCE are essentially the same kind of model, with a set of theoretical effects that can be parameterized by data.

Another issue is that in the section Statistical Analysis, the comparison of the QCE model to other variants of the QCE model is taken as robust support for the model. However, this comparison does not actually provide any support since the simpler models are missing some of the QCE elements and therefore are nearly guaranteed to provide a worse fit. It is useful to demonstrate which factors produce a worse fit, especially so that we can understand which of them are better for explaining the data, but it is inaccurate to claim that the worse fit provides support for QCE. In fact if the alternative versions of QCE produced a better fit than the full QCE it would imply only that the analysis was flawed because it cannot happen otherwise. The more effective comparisons are with the VP and NN models.

The style of writing uses mathematical terms imprecisely which is problematic especially in a computational paper.

E.g. in the introduction the phrase "exponentially larger" is used to compare datasets, but exponential refers to the shape of a curve, not merely the fact that something is much larger than something else.

The Results section is basically just the Methods section renamed to conform to the journal guidelines and Discussion section contains the results. The actual results should be moved into the results section.

Similarly in the supplemental it is written that an alternative model would need to be a

"a thousand times more detailed" but this is actually not true, as you could add subject-based variance into the model with only a few parameters, perhaps even only one.

Remove the phrase "under-development" when referring to the model. The reader understands this and it's unnecessary to keep repeating it.

It is difficult to appreciate the differences in Figure 2 given that the QCE model is not described until Figure 3.

I appreciate greatly that the effects of parameter removal have been reframed in terms of effect size. I am frankly surprised that the effect size differences in Figure 4 are as large as they are. It would be helpful to the reader to visualize how some of these differences manifest, perhaps in the supplementary. e.g. using the convention of Figure 2, how visibly different are the fits produced by the QCE model vs the QCE model where the "two categories" component is excluded?

(Remarks on code availability)

Reviewer #2

(Remarks to the Author)

The author has done an admirable job of revising the manuscript and adequately addressed the concerns I raised with the original submission.

(Remarks on code availability)

I performed a brief review of the code and have the following notes:

- The directory with the Python file should include a requirements.txt file specify versions of the dependencies (NumPy, PyTorch).

- There is no README describing what each file corresponds to, though it's easy enough to figure out.

The code generally looks good, but I have not executed the files.

Reviewer #3

(Remarks to the Author)

The author has answered all my questions and implemented my advice and/or added further information to address my minor concerns exceptionally well. I also read the comments from the other reviewers and am still highly enthusiastic about this study and the general approach. Some aspects of the model or their interpretations will certainly be proven wrong, further simplifications might be warranted, and other adaptations will be needed to account for data patterns that are ignored here. Nevertheless, in my opinion, this is a significant step into the right direction and provides a great starting point for those further explorations; or in the author's words: "the current QCE-VWM model, while being the best option available at this moment, is tentative and may be replaced by superior alternatives in subsequent iterations." (p. 12)

Signed. Heinrich Liesefeld

Thank you for adding Table 1 according to my request! I checked the references given and found them appropriate. Here are a few additional suggestions:

Regarding the trade-off between the quantity and quality of representations, I'd suggest to additionally cite Ye et al. (2019)

Regarding "Better-attended items are more likely to be remembered", I'd suggest to additionally cite Constant & Liesefeld (2021) for bottom-up attention, Emrich et al. (2017) for top-down attention, and Balaban et al. (2019) for the default spatial distribution of attention.

Regarding "The truncated normal distribution is superior to the von Mises distribution.", I'd suggest to additionally cite Oberauer (2023) who compares a von Mises to a Laplace distribution.

Balaban, H., Fukuda, K., & Luria, R. (2019). What can half a million change detection trials tell us about visual working memory? *Cognition*, 191, 103984. <https://doi.org/10.1016/j.cognition.2019.05.021>

Constant, M., & Liesefeld, H. R. (2021). Massive effects of saliency on information processing in visual working memory. *Psychological Science*, 0956797620975785. <https://doi.org/10.1177/0956797620975785>

Emrich, S. M., Lockhart, H. A., & Al-Aidroos, N. (2017). Attention mediates the flexible allocation of visual working memory resources. *Journal of Experimental Psychology. Human Perception and Performance*, 43(7), 1454–1465.

<https://doi.org/10.1037/xhp0000398>

Oberauer, K. (2023). Measurement models for visual working memory-A factorial model comparison. *Psychological Review*, 130(3), 841–852. <https://doi.org/10.1037/rev0000328>

Ye, C., Sun, H.-J., Xu, Q., Liang, T., Zhang, Y., & Liu, Q. (2019). Working memory capacity affects trade-off between quality and quantity only when stimulus exposure duration is sufficient: Evidence for the two-phase model. *Scientific Reports*, 9(1), Article 1. <https://doi.org/10.1038/s41598-019-44998-3>

I carefully re-read the paper and supplement and found only very minor issues that could be improved in the paper:

p. 5: spell out VP-F-NT

p. 6: I assume the "but" should be an "and" in this newly added sentence: "As illustrated in Figures 5a and 5b, the influence of one item on another is described by a normal function of the color difference between the two items in the effect on retention, but by a Mexican-hat-like function in the effect on bias."

p. 7: "In the Matthew effect, the smaller a category's weight is, the more it gets further reduced, hence the term "Matthew effect"". requires further explanation, maybe using the popular adage "the rich get richer and the poor get poorer" – not every reader is familiar with the original "Matthew effect"

p. 7: "is" should be "are" in "This implies that atypical colors, falling between the primary color categories, is generally at a disadvantage"

Fig. 6a: some of the text lines seem to be overlapping, cutting off part of the letters

(Remarks on code availability)

Version 2:

Reviewer comments:

Reviewer #1

(Remarks to the Author)

Thank you for addressing some of my concerns. I understand your points for the remaining items that you could not address.

(Remarks on code availability)

Reviewer #2

(Remarks to the Author)

The author did an admirable job of revising the manuscript in response to my first round of review and has now adequately addressed the questions I had on reviewing the code.

(Remarks on code availability)

I have not executed the code, but determined that it has what is needed to rerun the analyses.

A list of the main changes

- In response to reviewer 1's comment, the original AIC-based analysis has been replaced with an index that does not scale with the amount of data (Cohen's d , to measure the effect size of the comparisons). As detailed in the paper, these effect sizes are fairly large.
- When implementing the preceding change, a new criterion for adding/retaining mechanisms was adopted (complexity-adjusted $d > 0.2$, see details in Supplementary Information 6.1) because a Cohen's d of 0.2 is the usual standard for a "small but acceptable effect." As a result, a total of 17 of the initial 74 parameters have been dropped, leaving 57 parameters in the updated version of the QCE-VWM model. This streamlining has indeed made the model tighter and more elegant. Notably, the "mislocalization," which was considered a separate mechanism, has now been dropped.
- As part of the streamlining, the "red" and "red 2" categories from the last version are now modeled with a unified set of parameters. In addition, the color-wheel inhomogeneity (i.e., warm vs. cool colors) from the last version has also been merged into this red category.
- In response to reviewer 2's comment, a subset of the models from the van den Berg et al. (2014) factorial comparison paper has been replicated, and now serves as the baseline for the current work.
- An analogy of a puzzle game is now employed throughout the paper to clarify the main implications of this study. Previous research has identified numerous puzzle pieces. The significant advancement of the current study lies not in identifying additional puzzle pieces but in determining how the existing pieces can be assembled into an overall picture.
- In relation to the preceding point, Table 1 has been added, which lists 20 of the most important specific findings of this study and links them to particular previous studies in the literature.
- In response to reviewer 3's comment, a detailed explanation has been provided on how the scientific regret minimization method was applied. Additionally, a thorough explanation of the iterative refinement of the model has also been given.
- In response to reviewers 2 and 3's comments, two new mechanisms (Matthew effect and crosstalk) are described in greater detail, including explanations on how these effects differ from and relate to previous studies and how they were discovered.
- The effect of spatial inhomogeneity is now tentatively interpreted as an effect of spatial attention because it is consistent with the usual unequal distribution of attention acquired from our reading experience.

Reviewer 1:

This paper uses a massive data set (10 million trials and 40 million responses) to fit and test models. I think this approach is very intriguing and I also value the development of theoretical models as is done here. The author provides a comprehensive guide to how to replicate these analyses, although I did not replicate them myself. All of the files and data are available on OSF which is also commendable.

Response:

Thank you very much for your encouragement.

My biggest concern with this paper is that it is not clear what I have learned at the end of reading it. As there is no clear cut theoretical outcome. It is clear that a massive neural network does an overall better job of fitting the results in terms of absolute fit, but the argument is that the 74 parameter model is a better bargain because of the reduced complexity. That's all well and good, but what am I, a cognitive neuroscientist supposed to take from this finding that is going to affect my understanding of working memory?

Response:

Thank you very much for this comment. It is very helpful because future readers may have similar concerns, which can now hopefully be avoided.

Using a puzzle analogy, although numerous previous studies of visual working memory have identified many relevant mechanisms and factors, the primary takeaway from this study is that these puzzle pieces can be assembled to form a nearly complete overall picture. By saying "complete", I mean that the QCE-VWM model, as an integrative framework, has covered nearly all, if not all, of the underlying mechanisms. This completeness is supported by the fact that the QCE-VWM model is not merely a "good bargain"; it has outperformed the benchmark neural network.

I agree that this message of "putting puzzle pieces together" may be unfamiliar to researchers in this area, as it is not the usual perspective taken. However, I hope you will agree that it represents an important task that needs to be addressed.

In addition, the "overall picture" includes numerous specific findings. For 10 examples:

- 1. The interactions between items and chunking are based on pre-categorical, not category-based, color information.*
- 2. Consistent with the spirit of the slot model, only the retention rates, not the precision, are affected by other factors (i.e., interactions between items and spatial attention).*
- 3. Better-chunked patterns have a lower chance of being swapped, and are less attracted toward category centers, but are no more likely to be remembered.*
- 4. The relationship between the VWM precision of colors and the corresponding color categories is (mostly) multiplicative, not additive as the Bayesian principle would have suggested.*
- 5. Reddish colors are represented more precisely than other colors in category-biased component (i.e., red 2 category), but are less precisely than other colors in the unbiased component.*

6. *The weights of items affect each other, as revealed by the Matthew effect & Crosstalk.*

7. *Although widely used, the von Mises distribution is not as effective as the truncated normal distribution.*

8. *Spatial binding errors occur at the representation stage, but not at the response stage.*

9. *More typical colors, as defined by the categories, are more likely to be remembered.*

10. *Better-attended items are more likely to be remembered, their color categories are narrower and taller, and less effective at attracting the color-category biased component.*

There are too many of these findings to be fully listed. In the paper, the 20 most important findings are listed in Table 1. I must admit that none of these individual findings alone is substantial, but I hope you will agree that the collective value of all of them is quite considerable.

Another concern with the results is that the relative difference between the preferred model and the alternative/simplified models might be extremely small, and the statistical significance observed is a product of the large amount of data. However as a reader it is hard for me to understand the significance of the results, both in terms of statistical significance, and the magnitude of the effect. The critical issue is whether these specific qualitative features are important, because there are so many different parameters and in the QCE-VWM model. It is possible that with a model this complex, the actual components aren't that important, and nearly any combination of encoding mechanisms could be adjusted to match human performance similarly well. The author attempts to address with the alternative model analysis but I was unable to understand what these analyses really implied.

To explain, with data sets this large, it is hard to have a good intuition about what an AIC of e.g. 20,000 means, since AIC scales with the number of data points. 20,000 in one experiment might indicate an enormous loss in fit because the num of data points is fairly low, while in another experiment with more data, like this one, a tiny loss in the quality of the fit might produce the same massive AIC difference. A better approach would seem to be something that quantifies how much fit is lost with a measure that does not scale with the number of data points, such as % variance explained, or root mean squared error. You could then boot strap distributions of RMSEs for multiple fits of intact vs degraded models. The paper also uses NLLsr, but I am similarly unable to intuit what this means. For example, what does a drop from 5.08 to 5.075 imply about the quality of fit? This is a paper that I presume is targetted at the visual cognition community and its results should be presented in a way that are relatively straightforward to understand if it is to have an impact.

Response:

Thank you very much for this comment. I believe it has greatly improved the clarity of the paper.

Specifically, I have now replaced the AIC-based analysis with effect sizes. To demonstrate the QCE-VWM model's advantage over an alternative model across the 10,000 patterns, a t-test is employed. It goes without saying that the p-values are all extremely small. As you have pointed out, a more informative index would be a measure that does not scale with the number of data points. For this purpose, the effect size (Cohen's d) is perfect, so it is now utilized as the main criterion for model comparisons (please see Supplementary Information 8.3 and Supplementary Table 3). Specifically, I

have also developed a “complexity-adjusted d ” (please see Supplementary Information 6.1 for details on this index) to ensure that the benefit of each parameter is at least equivalent to a Cohen’s d of 0.2.

After implementing this new criterion (Cohen’s $d > 0.2$), a total of 17 of the initial 74 parameters have been dropped, leaving 57 parameters in the updated version of the QCE-VWM model. This streamlining has indeed made the model tighter and more elegant, so thank you very much for having suggested me to do this.

Some may point out that this criterion (Cohen’s $d > 0.2$) is not very stringent. Indeed, the effect sizes of some mechanisms are only slightly above 0.2, which is usually considered a “small effect.” However, it’s important to note that Cohen’s d is typically used in situations where experiments are tailor-made to highlight one specific mechanism or factor, unlike the randomly generated patterns in this study. Considering this, it seems fair to conclude that even an effect size with a Cohen’s d of 0.2 is decent in magnitude.

It is strange that order of report is not analyzed. It is well known that earlier reports have higher accuracy than later reports, and this is especially true when participants are allowed to select the order for report. This seems like a critical variable because the retrieval of the first vs the 4th color may differ in terms of what qualitative features are present.

c.f. Adam, K. C., Vogel, E. K., & Awh, E. (2017). Clear evidence for item limits in visual working memory. *Cognitive psychology*, 97, 79-97.

Response:

I agree that the order of reporting can significantly affect working memory, with the earlier items being remembered much better than the later ones. There is a reason why this aspect is not included in the QCE-VWM model. The modeling in the current study has been limited to the pattern-level summary (i.e., distribution of responses), overlooking trial-level details. Consequently, this effect becomes irrelevant in the pattern-level summary because it is averaged out.

The reason for focusing on pattern-level summarization is practical. Developing the current QCE-VWM model already required approximately eight months. Incorporating trial-level details into a model would necessitate information that is a thousand times more detailed and would significantly extend the optimization process. Moreover, it would introduce additional variables, leading to an increased number of potential models. Given these factors, developing a comprehensive trial-level model would be logistically impractical.

This issue of pattern-level summarization has been briefly addressed on page 4 and is further elaborated upon in Supplementary Information 3.1.

Of course, this practical limitation merely means that the “order of report” cannot be incorporated into the QCE-VWM model. However, it can be analyzed as a separate factor, and this has now been examined in Supplementary Information 9.11. Consistent with the findings of Adam, Vogel, & Awh (2017), later reports are significantly worse than earlier ones, primarily due to a decline in retention rate (i.e., item limits).

Fig 8b is misleading as the scale suggests that the difference between 2 and 74 parameters is pretty small, but this isn't true. The difference between 2 and 74 parameter models is enormous in terms of flexibility. This should perhaps be a log scale to allow us to see the difference between the smaller numbers

I suspect you could trim down the neural network substantially and keep a similar fit, this analysis should be attempted. For example, does a greatly reduced parameter version of a neural network achieve similar performance as the 74 parameters QCE, maybe 200 parameters or so? It would take some engineering to get the number of parameters that small given the number of inputs and outputs, but I suspect you could get the num output nodes down to 8 also.

Response:

The use of a log-scale here is indeed more appropriate. Thank you for this suggestion. I have now changed the x-axis of Figure 8b to a log-scale.

I have also trimmed down the neural network as you suggested, and the results are presented in Figure 8b. Specifically, four reduced versions of the guidance neural network are indicated by the gray dots. Their performance substantially decreases when the number of parameters is reduced to 628 and 208. This confirms that the neural network cannot remain effective without its complexity, highlighting the distinct advantage of the QCE-VWM model in simultaneously achieving effectiveness and parsimony.

Also I was not clear why the sin/cosine of the colors were used as inputs. The sin/cosine method of input doubles the number of input nodes and perhaps only adds confusion to the model. For example, the cosine is equal at 90 and 270 degrees but those colors are not more similar than the colors at 90 and 180 degrees are. So the input transform creates an inferential bias in color similarity that the model has to learn to overcome. A better solution might to put the colors on a grid and use the x/y coordinates of the color on the grid. This would ensure that colors close to each other in both x and y are more similar colors. However maybe this doesn't greatly affect the results but I had a hard time understanding the rationale.

Response:

Sorry for causing confusion. However, I actually meant precisely what you suggested: using the x/y coordinates of the color on a color ring. The cosine and sine functions are used to calculate the x/y coordinates. When converting polar coordinates to Cartesian coordinates, all the dots on a ring (assuming a radius of 1 for simplicity) can be calculated as $(x,y)=[\cos(\text{angle}),\sin(\text{angle})]$. This has now been clarified.

I could not understand the generalization rate analysis of S8.4. What does 100.5% mean and how was it computed?

Response:

This term has now been renamed "generalizability ratio" for better clarity. It should also be clarified that, although its spirit remains the same, the calculation of this generalizability ratio has been

changed due to your comment above. Previously, it was calculated based on NLLsr, but it is now based on the Cohen's d values. Specifically, the generalizability ratio is now defined as the ratio between the "Cohen's d value for the validation set" and the "Cohen's d value for the training set." A ratio of 100% indicates that the advantage is perfectly generalizable, whereas a ratio of 0% indicates that the advantage is completely artificial and non-generalizable. The average generalizability ratio is now 99.1% (refer to Supplementary Table 4 for detailed values), suggesting that it is perfectly generalizable.

On a technical note, in the AIC comparison, if one of the features of the model is set to 0, how many parameters does that remove from the total, and is that change reflected in the AIC?

Response:

This is no longer relevant because AIC is no longer used, and now the analysis is based on Cohen's d values due to your comment above. However, to answer your question, yes, the AIC in the previous version did take into account the change in the number of parameters.

In the Preliminary analysis section: "with an average Root Mean Square Error (RMSE) of 47° on the color wheel." I could not understand how the error was being computed. Was this the error of participant responses for all four responses relative to their actual colors?

Response:

Yes, precisely, these are the errors of participants' responses for all four responses relative to their actual colors. This has now been clarified on page 14.

I could not understand exactly what the dashed line represented in supplemental Figure 1.

Response:

The dashed line represents the NLLsr of the QCE-VWM model, provided for easy comparison between QCE-VWM and neural networks. Specifically, it demonstrates that QCE-VWM has outperformed all neural networks (i.e., their performance on validation sets).

Reviewer 2:

In this manuscript, the author reports the results of a massive experiment to understanding visual working memory (VWM), building upon the author's prior work in applying the scientific regret minimization technique to visuospatial memory. After collecting a new benchmark dataset of 40 million behavioral responses over 10,000 color patterns, the author develops a new model of VWM, termed QCE-VWM. The model, which is simultaneously less complex (and more interpretable) than traditional neural network models while avoiding the often squishy verbal nature of model specification in this domain, surpasses them in data fitting, showcasing the effectiveness of the scientific regret minimization method in this domain. By integrating concepts drawn from both slot and resource models, the study attempts to reconcile conflicting theories in VWM. It also introduces new mechanisms like the Matthew effect and crosstalk.

In general, I find the work promising and the dataset alone would be enough for the paper to be a valuable resource to the community.

Response:

Thank you very much for your encouragement.

That being said, I have several major concerns about the work in its current form.

1. The Matthew effect and the crosstalk effect are each defined in only a single sentence in the main text, which is too little justification for what even the abstract claims as a contribution to the literature with respect to novel working memory mechanisms. The supplement goes into more detail about their respective definitions, but provides no motivation in terms of how these effects relate to what is known, nor how these effects were discovered.

Response:

Thank you very much for the comments. One reason for the inadequate explanation of the Matthew effect and the crosstalk effect was that they emerged as outcomes of open-ended exploration, rather than being motivated by a specific theoretical hypothesis. You are correct in pointing out that readers may be interested in knowing more about these mechanisms. So I have now clearly explained their relevance to what is already known.

Specifically, we know the importance of color categories, and we also recognize that items can influence each other. So I tried to ask the "conjunction question" of how the color category weights of items affect each other, and found the Matthew effect and the crosstalk. This explanation is now provided on page 7, where these two mechanisms are first introduced. Furthermore, in the discussion section on page 10 (see also Supplementary information 6.4), I have discussed all the "conjunction questions" between the 3 topics: color categories, the interactions between items, and random guesses.

Additionally, I have detailed how these effects were discovered in Supplementary Information 6.4. Essentially, these separate effects were identified through open-ended optimization.

I hope the explanations are now sufficiently clear. I understand that some may be disappointed that these discoveries were made through exploration rather than being theory driven. After 20 years of active research on VWM, with all the low-hanging fruits already picked, it is perhaps natural that the remaining mechanisms are not conceptually straightforward and must be identified in this manner.

In any case, after your prompt to clarify, I now agree it is important to make this aspect clear to readers, enhancing their understanding of the exploratory nature of this study. So, thank you for raising this point.

2. The link between the proposed model's design space and those of extant models in the literature is insufficiently clear. I would have strongly preferred a model comparison like that in the van den Berg et al. (2014) factorial comparison paper, Fig. 3c, where the proposed model, all the ablated versions, as well as all the major extant models from the literature are compared head to head. My guess is that they will perform quite well for a narrow slice of the stimulus space, and quite poorly for random samples of the stimulus space. Taking this approach has the dual function of showing the community that the work has taken previous formal models in this domain seriously, while also showing the community that those previous models fail when considered at scale.

Response:

This is a great comment. Comparing our work to van den Berg et al. (2014) and their factorial comparison paper does indeed make the present study empirically more solid and theoretically more well-founded. Thank you very much.

On the theoretical side, the present study shares a common goal with the factorial comparison by van den Berg et al. (2014), namely, the simultaneous testing of multiple factors. However, there is also a critical difference. Unlike the factorial comparison study, which conducted exhaustive testing within a predefined space of solutions, the present study adopts a data-driven iterative search within an unlimited space of solutions. Making this comparison will certainly help readers understand the approach of the current study more precisely.

On the empirical side, I have attempted to replicate the models from van den Berg et al. (2014) factorial comparison paper. Some of these theoretical options are inapplicable here due to the nature of the stimuli and analysis (please see Supplementary Information 5 for details), resulting in a total of $2 \times 2 \times 2 = 8$ models. Consistent with the van den Berg et al. (2014) study, the VP-F-NT model emerged as the best among these 8 and is now used as the new "baseline" for the present study. Following your suggestion, the QCE-VWM model, along with the 17 alternative models, the neural networks, and the 8 models from the factorial comparison analysis, are all compared in Figure 9.

While I fully understand the value of an exhaustive testing, it is impractical to test "all" the ablated versions of the QCE-VWM model. The current QCE-VWM model has 57 parameters, and exhaustively testing all possibilities for dropping a subset of these parameters would lead to $2^{57} = 1.4E+17$ models. Even considering just the 17 aspects of the 17 alternative models, we would still have $2^{17} = 131,072$ models. In a way, the 17 alternative models represent a selective subset of all the ablated versions: the "immediately conceivable alternatives." I hope you will agree that this list is reasonably sufficient. But

of course, if there is any specific ablated model (or any specific extant model) you would like me to test, I would be very happy to include them in the next version.

3. It is unclear what the inclusion/exclusion criteria on working memory mechanisms are that justify an 18-feature design space of models. Many of these model components are indeed well-supported by the literature, but there are other components that might have been included but were not (e.g., positional effects, kurtosis from cross-participant or cross-trial variability in precision, time-of-day effects, serial dependence, interference across trials in a run, interactions between iconic and working memory, display resolution, participant-level features like gender, etc.). The paper includes a bit of hedging to address this (e.g., if anything is missing, a new model with the added component could be tested and might surpass the present model), but that doesn't help address the specific question of what's in vs. out for the present work. Was some kind of systematic literature review completed?

Response:

I strive to include any factor that could help improve the model. Thus, the factors you mentioned were not excluded due to any conceptual-level inclusion/exclusion criteria but for a practical reason. Specifically, the modeling in the current study has been limited to a pattern-level summary (i.e., distribution of responses), overlooking trial-level details. As a result, these effects become irrelevant in the pattern-level summary because they are averaged out.

The reason for focusing on pattern-level summarization is practical. Developing the current QCE-VWM model already required approximately eight months. Incorporating trial-level details into a model would necessitate information that is a thousand times more detailed and would significantly extend the optimization process. Moreover, it would introduce additional variables, leading to an increased number of potential models. Given these factors, developing a comprehensive trial-level model would be logistically impractical.

This issue of pattern-level summarization has been briefly addressed on page 4 and is further elaborated upon in Supplementary Information 3.1.

Of course, this practical limitation merely means that these effects cannot be incorporated into the QCE-VWM model. However, they can be analyzed as separate factors. The positional effects, as well as the effects of age and gender, have now been examined respectively in Supplementary Information sections 9.11 and 9.12. By "positional effects," I assume you mean "order of report," since the role of spatial location has already been incorporated into the model.

The other factors have been skipped for various reasons. We did not collect the information about time-of-day and display resolution, so they cannot be analyzed. The interactions between iconic and working memory cannot be analyzed because we have not manipulated the retention interval. The variability in precision was already part of the model, as reflected by the low-precision component in step 3f. I did try to analyze the interference across trials in a run (e.g., serial dependence) but found there is not much of anything worth mentioning. This is probably because we always used four colors, which exhausted the VWM mechanisms and left little room for the affected items to be impacted. This also diluted the effects of each affecting item.

- Apologies if I am being dense, but I did not catch what the QCE acronym in the model name expands to.

Response:

The QCE stands for a quasi-comprehensive exploration model. “Comprehensive exploration” refers to the current approach, which is now more clearly explained throughout the paper, with “quasi” indicating an acknowledgment of potential incompleteness. This explanation now appears at the beginning of the section titled “A comprehensive exploration model” on page 5.

- The supplemental materials are thorough and do an excellent job of providing the information needed to replicate the work.

Response:

Thank you.

Reviewer 3:

This massively large-scale study employs a whole-report color visual-working-memory task and models the results with a convolutional neural network to estimate how well the data can be explained by a flexible data-driven model. On each trial of this task, administered on participants' own devices (most likely mobile phones) via the multi-purpose personal app WeChat, participants saw four colors and subsequently had to report all four colors in a self-determined order on a color wheel. The main achievement of the current study is the development of a theoretically derived framework model that can account for the data as good or even better than the neural network. Based on the full model the necessity of its various theoretically derived component mechanisms is evaluated. It turns out that even mechanisms that were previously considered competitors for explaining the same data patterns all explain unique portions of the variance.

This is a highly interesting novel approach to cognitive psychology that I was not aware of before and I will definitely follow the author's work with this approach closely from now on. The sheer amount of data collected is impressive, although a tradeoff with regard to data quality had to be made (see in particular Point 1 below). The modeling takes a large number of known influences on VWM performance into account (chunking, spatial biases, color categories, attraction/repulsion, swapping, quantity/quality tradeoff, guessing...) and also considers additional influences (Matthew and Crosstalk effect). It is therefore also far more extensive than any previous modeling in this area. The exploration of model versions documented mainly in the supplement seems appropriate and theoretically well informed. I've read all materials with great interest and believe that this work will have a profound and persisting influence on my own thoughts on VWM. This work is useful for the large number of new specific and testable hypotheses alone, but it provides so much more. I also found the information on the author's webpage on the development of this approach highly interesting and recommend reading it as well: <https://sites.google.com/view/liqianghuang/story?authuser=0>

Signed. Heinrich Liesefeld

Response:

Thank you very much for your thoughtful feedback. It's incredibly motivating to see my work being understood and valued by peers in the field. Your approval will certainly encourage us to push further in this direction. We're grateful for your support and keen interest.

Here are my minor comments that hopefully help to improve the manuscript even further:

1. There should be a disclaimer in the supplement and paper that the colors were **not** CIE-L*a*b, because the screens were not calibrated; I assume that colors were L*a*b* under the unlikely assumption that screens were calibrated to sRGB. Also note that variation in colors between devices likely contributed to interindividual variance in performance and to complex distortions of the color space. Could this inaccuracy in color presentation account for the deviation from the Bayesian predictions (e.g., Suppl. Fig. 3)?

Response:

Thank you for reminding me to clarify this. This point has now been briefly mentioned in the Methods section (see the beginning of page 14) and elaborated upon in Supplementary Information 2.2. It is challenging to precisely assess the impact of this issue. As an indirect assessment, I have examined the overall distribution of responses on the color wheel, which is now presented in Supplementary Figure 1. The peaks are very tall, and their amplitudes are comparable to those in a previous study (see Fig 6b of Bae et al., 2015). Therefore, I believe the participants in this study perceived the colors reasonably consistently. The relatively minor inconsistencies are unlikely to have significantly impacted the results. Had there been significant impacts, the peaks would differ from participant to participant and would be smoothed out.

2. The author strikes a great balance between detail and brevity in his writing. I only found a few aspects that were not yet sufficiently explained for my taste:

2a. I think more detailed explanation of the “scientific regret minimization” approach is needed for the reader to understand this paper. It is partially given in the section “Neural network as the target for model development” (p. 4), but some more elaboration and clearer indication which of these ideas are at the core of scientific regret minimization and which have been added for the current purposes would be helpful, I feel.

Response:

Thank you for reminding me to clarify this. It is very helpful because future readers may have similar concerns, which can now hopefully be avoided.

More discussions have now been provided in pages 4-5, and also elaborated upon in Supplementary Information 6.3. In brief, the “scientific regret minimization” method utilizes the predictions produced by the neural network as guidance for the development of QCE-VWM model. First, the predictions of an under-development conceptual model are compared with those of the guidance neural network to gain insights into what is missing in the former. Second, the predictions of the guidance neural network are used as virtual data, allowing us to go beyond the 10,000 patterns for which we actually have data and explore all $360^4 = 16,796,160,000$ possible patterns.

I have also provided some details of how these are done. For example, to understand how the precision and bias of an item’s distribution of responses are affected by other items, it would be ideal to have data for all possible patterns. This would allow us to directly observe how an item’s precision and bias change when another item shifts on the color wheel. However, this is impossible on the actual data because we can only find scattered examples and are left to infer what might happen in the gaps between these examples. But now, with this “scientific regret minimization”, we can use virtual data to do this and form some insights on what’s the underlying mechanisms.

I was somewhat confused by the fact that the approach is introduced in these terms and then there is a statement that the model did “surpass the target neural network in terms of data fit” (p. 7) – how and in which way can it surpass the neural network model if the neural network model was the target? Some

confusion needs to be resolved here. “the QCE-VWM has made considerable progress, outperforming the target neural network” (p. 10) indicates that I misunderstood what “target” means here.

Response:

Thank you for reminding me to clarify this issue. My previous description might have been somewhat misleading. Essentially, the “scientific regret minimization” method is utilized only for gaining insights, while the model is always fitted to the actual data. This clarification has now been explicitly made on page 5. Moreover, the term “target neural network” used in the previous version may have contributed to the confusion, so it has been renamed to “guidance neural network.”

In the final part of Supplementary Information 6.3, I have also addressed the question: How is it possible for the QCE-VWM model to outperform the guidance neural network in terms of data fit, despite being influenced by it? The QCE-VWM model, though guided by the neural network, is always calibrated against the actual data. Therefore, it is technically possible for the outcome of this guidance to surpass the guidance itself. This is because neural networks, by their nature, ensure that their predictions are never perfect. They may come very close to the truth but are always slightly off from the exact truth. However, when used in conjunction with conceptual judgment, they may guide us to the exact truth.

2b. “ReLU activation function” is mentioned, but not explained.

Response:

Thanks for pointing this out. “ReLU(x) = max(x, 0)” has now been clarified on page 4.

2c. Given that the Matthew and Crosstalk effects are introduced to this community here, a more detailed explanation seems warranted on p. 6. This should contain a detailed explanation in how far these types of effects are not already covered by previously proposed VWM mechanisms.

Response:

Thank you very much for the comments. I have now clearly explained their relevance to what is already known. Specifically, we know the importance of color categories, and we also recognize that items can influence each other. Therefore, I sought to ask the “conjunction question” of how the color category weights of items affect each other. This explanation is now provided on page 7, where these two mechanisms are first introduced.

3. I believe that readers would appreciate core references for each of the introduced VWM mechanisms in the supplement (as far as these have been proposed before). What I have in mind here is that for each model component there could be a reference to 2-3 core papers that have suggested, empirically shown, or reviewed the respective cognitive/neuronal mechanism. If the author starts collecting these references, I’m happy to supplement them based on my knowledge of the literature during the next round of reviews.

Response:

This is an excellent suggestion. I have now created a table (Table 1, on page 27) that lists the relatively important specific findings, as well as the relevant references for each of them. If you could provide some suggestions for new references, that would be wonderful.

4. It is claimed that the conversion to cosine and sine values “indeed improves the predictions of the neural network.” (Suppl., p. 3) – but compared to which alternative measure of color does it improve predictions? I can see that the apparent dissimilarity between, say, 258° and 2° on a linear scale would cause trouble for the model, but is there really no proper solution for handling circular spaces using just a single value/input neuron?

Response:

In the Supplementary Information 4, it has now been clarified that the comparison was made with “another neural network that directly uses angles to represent colors.” I have tried to look for other options, but this is the most straightforward option I can find.

5. Instead of asking the reader “to use document comparison software to highlight the changes” (Suppl., p. 13), delta versions could be added to the OSF project.

Response:

Thanks for this suggestion, I have now created these documents (Model_16_changes.docx, Model_17_changes.docx) and added them to the online materials.

Reviewer 1:

I think this version of the paper is a marked improvement over the previous one but I still think it could be substantially improved by some additional revisions.

On this second review I have now come to understand its value, which does not come out very clearly in its current form. Essentially this paper is a comparison of the effectiveness of the accumulated explanatory power of cognitive science to explain working memory against a theory-free neural network. The value of theory has always been ascribed as providing concise (i.e. low parameter) descriptions of natural phenomena (hence the phrase “to carve nature at its joints”), and the demonstration here is indeed that the accumulated wisdom of decades of work on WM does indeed provide a better explanation of empirical observations than a massive neural network, despite the difference in parameters. This is a valuable and reassuring demonstration of the effectiveness of good theory.

However this framing is somewhat obscured in the introduction which connects the QCE approach to Artificial Intelligence given that a large amount of data is used to parameterize the model. Just because an approach is driven by large data sets, that does not mean it is AI. One could even argue that the QCE approach here seems more like the opposite of AI as it is currently construed in the literature, given that it is using theory derived from experimental work to form the basis of the candidate model rather than just fitting a massive set of parameters (which is what the neural network is doing).

Response:

I never intended to imply that the QCE model is an AI model. Instead, I meant to convey that AI is used as tools to achieve the goals of experimental psychology. This has been made clearer in the introduction section with the following statement (see page 4): "As mentioned above, AI is used as a tool in this study to achieve the goal of experimental psychology (theoretical insights). Therefore, it's important to note that this targeted model is not an AI model but, like previous models in the VWM field, a theory-based model."

You have articulated the main strength of this study very well: the accumulated wisdom of decades of work on WM indeed provides a better explanation of empirical observations than a massive neural network. I have tried to make the same point, but you have expressed it more clearly. This statement has now been added to the relevant section on page 10. Thank you very much for this.

This is not my paper of course and these are opinions about framing, rather than observations of factual errors, so the author and editor may disregard.

Response:

Thank you very much for your understanding.

Another issue is that the framing of the results section could be improved. I would start off by making it clear that there are three models that will be compared, and explain why these models are being used. In the current form the reader has to puzzle out that there are three different models and the

terminology is confusing. e.g. the neural network is a guidance model and the VP model is the baseline. Why these terms? I don't understand. To me, the neural network is a baseline of how good the fit can get without theory. The VP is an alternative theoretical model to QCE. VP and QCE are essentially the same kind of model, with a set of theoretical effects that can be parameterized by data.

Response:

The mission of this study is the development of the QCE model through data-driven comprehensive exploration. The baseline model (VP in this case) represents what has been achieved through traditional theory-driven model development, making it the starting point of model development, hence the name "baseline." The guidance neural network provides a tentative model for what a comprehensive model should explain and is used to guide the development of the QCE model, hence the name "guidance."

This has now been made clear with a new paragraph under the heading "The overall plan" (see page 4). I want to thank you for reminding me to clarify this. I believe this will help many other readers who may have similar confusions.

Another issue is that in the section Statistical Analysis, the comparison of the QCE model to other variants of the QCE model is taken as robust support for the model. However, this comparison does not actually provide any support since the simpler models are missing some of the QCE elements and therefore are nearly guaranteed to provide a worse fit. It is useful to demonstrate which factors produce a worse fit, especially so that we can understand which of them are better for explaining the data, but it is inaccurate to claim that the worse fit provides support for QCE. In fact if the alternative versions of QCE produced a better fit than the full QCE it would imply only that the analysis was flawed because it cannot happen otherwise. The more effective comparisons are with the VP and NN models.

Response:

I think there is some misunderstanding here. It is indeed true that adding an extra factor or mechanism will always improve the model, and this is true for model comparison in general. Then, statistical indexes are used to assess whether the magnitude of that improvement is large enough to justify the extra factor or mechanism. In usual model comparison, AIC and BIC are used for that purpose, and CAD (complexity-adjusted Cohen's d) is now used in this study according to your earlier suggestions. Therefore, I believe it is accurate to claim that these statistical tests demonstrate that the QCE is better than the alternative models in the sense that the magnitude of improvement exceeds a given statistical criterion.

To be clear, when a factor is taken away in an alternative model, the CAD between QCE and that alternative model will always be positive, but it will not always be greater than 0.2. It is indeed the case that one mechanism of the earlier version of this study (mis-localization) was removed for this reason.

The style of writing uses mathematical terms imprecisely which is problematic especially in a computational paper.

E.g. in the introduction the phrase “exponentially larger” is used to compare datasets, but exponential refers to the shape of a curve, not merely the fact that something is much larger than something else.

Response:

Thank you. This has been corrected as suggested.

The Results section is basically just the Methods section renamed to conform to the journal guidelines and Discussion section contains the results. The actual results should be moved into the results section.

Response:

Thank you for the suggestion; you certainly have a point. However, there is a difficult situation here because those parts that may be more suitable for results (e.g., the comparison in Figure 8b and Figure 9) become clearer after some discussions are made (e.g., the “putting puzzle pieces together” metaphor). Therefore, I believe this is the way that flows the most smoothly. I hope you will be OK with this.

Similarly in the supplemental it is written that an alternative model would need to be “a thousand times more detailed” but this is actually not true, as you could add subject-based variance into the model with only a few parameters, perhaps even only one.

Response:

I was not referring to the complexity of the model, but to the amount of data that needs to be handled. This has now been rephrased as follows: “A model at the trial level would need to handle data in much more detail and would take significantly longer to optimize.”

Remove the phrase “under-development” when referring to the model. The reader understands this and it’s unnecessary to keep repeating it.

Response:

Thank you. This has been corrected as suggested. The term “under-development” is used only once to clarify this issue and is omitted later.

It is difficult to appreciate the differences in Figure 2 given that the QCE model is not described until Figure 3.

Response:

I understand the concern, but there is no perfect solution. I want to show the readers some examples of distributions at this point, and it is best to present the distributions along with the models' predictions. But I agree with you that this may be potentially confusing, so I have made it clear in the figure legend that these models will be explained below.

I appreciate greatly that the effects of parameter removal have been reframed in terms of effect size. I am frankly surprised that the effect size differences in Figure 4 are as large as they are. It would be helpful to the reader to visualize how some of these differences manifest, perhaps in the supplementary. e.g. using the convention of Figure 2, how visibly different are the fits produced by the QCE model vs the QCE model where the “two categories” component is excluded?

Response:

Thank you very much for the praise.

For the suggested visualization of the differences between models, I have conducted the recommended analysis. Below are two representative examples that visualize the differences between the QCE-VWM model (white curve) and the “eliminating two categories” model (black curve).

Although we can see that the white curves fit the data better than the black curve, it is difficult to grasp the conceptual meanings of these differences from just a couple of graphs. I believe this is partly because the individual-pattern data is quite noisy, making it challenging to determine precisely why the model is better.

For this reason, I am inclined to think that these figures are not informative enough to be included. I hope you will be OK with this.

Reviewer 2:

The author has done an admirable job of revising the manuscript and adequately addressed the concerns I raised with the original submission.

Response:

Thank you very much for the great comments provided earlier, which have helped me improve this paper.

Reviewer 2 (Remarks on code availability):

I performed a brief review of the code and have the following notes:

- The directory with the Python file should include a requirements.txt file specify versions of the dependencies (NumPy, PyTorch).

Response:

This has now been provided.

- There is no README describing what each file corresponds to, though it's easy enough to figure out.

Response:

This has now been provided.

The code generally looks good, but I have not executed the files.

Reviewer 3:

The author has answered all my questions and implemented my advice and/or added further information to address my minor concerns exceptionally well. I also read the comments from the other reviewers and am still highly enthusiastic about this study and the general approach. Some aspects of the model or their interpretations will certainly be proven wrong, further simplifications might be warranted, and other adaptations will be needed to account for data patterns that are ignored here. Nevertheless, in my opinion, this is a significant step into the right direction and provides a great starting point for those further explorations; or in the author's words: "the current QCE-VWM model, while being the best option available at this moment, is tentative and may be replaced by superior alternatives in subsequent iterations." (p. 12)

Signed. Heinrich Liesefeld

Response:

Thank you very much for the great comments provided earlier, which have helped me improve this paper.

Thank you for adding Table 1 according to my request! I checked the references given and found them appropriate. Here are a few additional suggestions:

Regarding the trade-off between the quantity and quality of representations, I'd suggest to additionally cite Ye et al. (2019)

Regarding "Better-attended items are more likely to be remembered", I'd suggest to additionally cite Constant & Liesefeld (2021) for bottom-up attention, Emrich et al. (2017) for top-down attention, and Balaban et al. (2019) for the default spatial distribution of attention.

Regarding "The truncated normal distribution is superior to the von Mises distribution.", I'd suggest to additionally cite Oberauer (2023) who compares a von Mises to a Laplace distribution.

Balaban, H., Fukuda, K., & Luria, R. (2019). What can half a million change detection trials tell us about visual working memory? *Cognition*, 191, 103984. <https://doi.org/10.1016/j.cognition.2019.05.021>

Constant, M., & Liesefeld, H. R. (2021). Massive effects of saliency on information processing in visual working memory. *Psychological Science*, 0956797620975785. <https://doi.org/10.1177/0956797620975785>

Emrich, S. M., Lockhart, H. A., & Al-Aidroos, N. (2017). Attention mediates the flexible allocation of visual working memory resources. *Journal of Experimental Psychology. Human Perception and Performance*, 43(7), 1454–1465. <https://doi.org/10.1037/xhp0000398>

Oberauer, K. (2023). Measurement models for visual working memory-A factorial model comparison. *Psychological Review*, 130(3), 841–852. <https://doi.org/10.1037/rev0000328>

Ye, C., Sun, H.-J., Xu, Q., Liang, T., Zhang, Y., & Liu, Q. (2019). Working memory capacity affects trade-off between quality and quantity only when stimulus exposure duration is sufficient: Evidence for the two-phase model. *Scientific Reports*, 9(1), Article 1. <https://doi.org/10.1038/s41598-019-44998-3>

Response:

Thank you very much for these great suggestions; I have added them to the paper.

I carefully re-read the paper and supplement and found only very minor issues that could be improved in the paper:

p. 5: spell out VP-F-NT

Response:

Thanks. This has now been briefly explained.

p. 6: I assume the “but” should be an “and” in this newly added sentence: “As illustrated in Figures 5a and 5b, the influence of one item on another is described by a normal function of the color difference between the two items in the effect on retention, but by a Mexican-hat-like function in the effect on bias.”

Response:

Thank you. This has been corrected as suggested.

p. 7: “In the Matthew effect, the smaller a category’s weight is, the more it gets further reduced, hence the term “Matthew effect””. requires further explanation, maybe using the popular adage “the rich get richer and the poor get poorer” – not every reader is familiar with the original “Matthew effect”

Response:

Thank you. This has been corrected as suggested.

p. 7: “is” should be “are” in “This implies that atypical colors, falling between the primary color categories, is generally at a disadvantage”

Response:

Thank you. This has been corrected as suggested.

Fig. 6a: some of the text lines seem to be overlapping, cutting off part of the letters.

Response:

This is an artifact of the PDF file due to image resolution; it displays normally when you zoom in. In any case, thank you very much for reminding me about this. I will keep an eye on it and make sure it displays normally in the final version.

In this round, the reviewers have not raised any concerns, as outlined below.

REVIEWERS' COMMENTS

Reviewer #1 (Remarks to the Author):

Thank you for addressing some of my concerns. I understand your points for the remaining items that you could not address.

Reviewer #2 (Remarks to the Author):

The author did an admirable job of revising the manuscript in response to my first round of review and has now adequately addressed the questions I had on reviewing the code.

Reviewer #2 (Remarks on code availability):

I have not executed the code, but determined that it has what is needed to rerun the analyses.